# HYMBA: A HYBRID-HEAD ARCHITECTURE FOR SMALL LANGUAGE MODELS

**Xin Dong**[1*], **Yonggan Fu**[1,2*], **Shizhe Diao**[1], **Wonmin Byeon**[1], **Zijia Chen**[1],
**Ameya Sunil Mahabaleshwarkar**[1], **Shih-Yang Liu**[1,3], **Matthijs Van Keirsbilck**[1],
**Min-Hung Chen**[1], **Yoshi Suhara**[1], **Yingyan (Celine) Lin**[1,2], **Jan Kautz**[1], **Pavlo Molchanov**[1]
[1]NVIDIA    [2]Georgia Institute of Technology    [3]Hong Kong University of Science and Technology

## ABSTRACT

We propose Hymba, a family of small language models featuring a hybrid-head parallel architecture that integrates attention mechanisms and state space models (SSMs) within the same layer, offering parallel and complementary processing of the same inputs. In this hybrid-head module, attention heads provide high-resolution recall, while SSM heads facilitate efficient context summarization. Additionally, we introduce learnable meta tokens, which are prepended to prompts to store critical meta information, guiding subsequent tokens and alleviating the "forced-to-attend" burden associated with attention mechanisms. Thanks to the global context summarized by SSMs, the attention heads in our model can be further optimized through cross-layer key-value (KV) sharing and a mix of global and local attention, resulting in a compact cache size without compromising accuracy. Notably, Hymba achieves state-of-the-art performance among small LMs: Our Hymba-1.5B-Base model surpasses all sub-2B public models and even outperforms Llama-3.2-3B, achieving 1.32% higher average accuracy, an $11.67\times$ reduction in cache size, and $3.49\times$ higher throughput.

**Models on Hugging Face:** Hymba-1.5B-Base | Hymba-1.5B-Instruct

## 1 INTRODUCTION

Transformers, with their attention-based architecture, have become the dominant choice for language models (LMs) due to their strong performance, parallelization capabilities, and long-term recall through key-value (KV) caches (Vaswani, 2017). However, their quadratic computational cost and high memory demands pose efficiency challenges. In contrast, state space models (SSMs) like Mamba (Gu & Dao, 2023) and Mamba-2 (Dao & Gu, 2024) offer linear complexity and efficient hardware optimization but struggle with memory recall tasks, affecting their performance on general benchmarks (Waleffe et al., 2024; Arora et al., 2024a). While existing hybrid models that stack attention and SSM layers have demonstrated potential (Lieber et al., 2024; Ren et al., 2024), they can introduce information bottlenecks when one layer type is not well-suited for specific tasks, requiring compensation from subsequent layers.

In light of this, we propose Hymba, a novel LM architecture that integrates attention heads and SSM heads within the same layer, offering parallel and complementary processing of the same inputs. This hybrid-head approach allows each layer to simultaneously harness both the high-resolution recall of attention and the efficient context summarization of SSMs, increasing the model's flexibility and expressiveness in handling various types of information flows and memory access patterns.

To further enhance the achievable performance of Hymba, we introduce learnable meta tokens that are prepended to the input sequences and interact with all subsequent tokens even in sliding window attention. These meta tokens appear to act as a compressed representation of world knowledge and alleviate the issue of "softmax attention not being able to attend to nothing" (Bondarenko et al., 2023; Miller; Xiao et al., 2023), improving performance across both general and recall-intensive tasks. In addition, inspired by findings in (Brandon et al., 2024) that consecutive layers have a high correlation in the KV cache, we propose sharing the KV cache between layers as well. Additionally,

---

*Equal contribution.

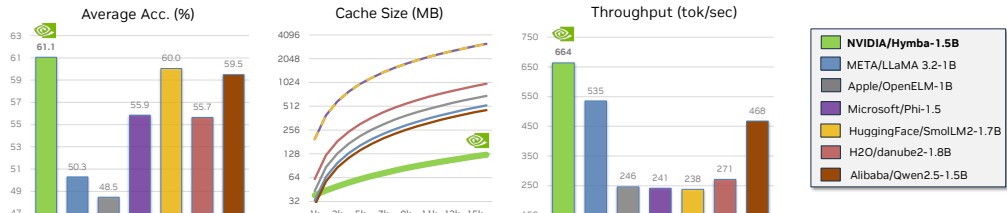

Figure 1: Performance comparison of Hymba-1.5B against sub-2B models in terms of average task accuracy, cache size (MB) relative to sequence length, and throughput (tok/sec). Specifically, the tasks include 5-shot MMLU, ARC-C, ARC-E, PIQA, Hellaswag, Winogrande, and SQuAD-C, and the throughput is measured on an NVIDIA A100 with a sequence length of 8k and a batch size of 128 using PyTorch. For models encountering out-of-memory (OOM) issues during throughput measurement, we halve the batch size until the OOM is resolved. This approach is used to measure the maximal achievable throughput without OOM.

we incorporate a mix of global and local (sliding window) attention, with the latter used in most of layers, to further reduce cache costs without compromising accuracy.

Comprehensive evaluations and ablation studies demonstrate that Hymba not only establishes new state-of-the-art (SOTA) benchmark performance across a wide range of tasks but also achieves greater efficiency compared to transformers and previous hybrid models. We provide the benchmark with other representative small LMs in Fig. 1, with more comprehensive benchmarks in Fig. 6. For instance, in commonsense reasoning tasks, Hymba-1.5B can outperform Llama-3.2-3B with 1.32% higher average accuracy, while requiring $11.67\times$ smaller cache size and being $3.49\times$ faster.

To optimize Hymba for on-device tasks, we further employ supervised finetuning and direct preference optimization (DPO) (Rafailov et al., 2024). Our instruction-tuned model, Hymba-1.5B-Instruct, achieves best-in-class performance on GSM8K, GPQA, and the Berkeley function-calling leaderboard, surpassing Llama-3.2-1B. Additionally, parameter-efficient finetuning shows Hymba's strong potential in this setting. For instance, a DoRA (Liu et al., 2024d)-finetuned version of Hymba-1.5B outperforms Llama3.1-8B-Instruct by 2.4% on RoleBench (Wang et al., 2023).

## 2  HYMBA: THE PROPOSED HYBRID-HEAD ARCHITECTURE

SSMs such as Mamba (Gu & Dao, 2023) were introduced to address the quadratic complexity and large inference-time KV cache issues of transformers. However, due to their low-resolution memory, SSMs struggle with memory recall and reasoning accuracy (Waleffe et al., 2024; Jelassi et al., 2024; Arora et al., 2024a). To overcome these limitations, we propose a roadmap for developing efficient and high-performing small LMs in Tab. 1 and outlined as follows:

**Fused hybrid-head modules.** Fusing attention and SSM heads in parallel within a hybrid-head module outperforms sequential stacking (see Tab. 1 (A)-(B) and Sec. 2.1). Both heads process the same information simultaneously, leading to improved reasoning and recall accuracy. We argue that sequential fusion lacks synergy, as both blocks operate on each set of inputs independently.

| Configuration | Commonsense Reasoning (%) | Recall (%) | Throughput (token/sec) | Cache Size (MB) | Design Reason |
|---|---|---|---|---|---|
| **Ablations on 300M model size and 100B training tokens** | | | | | |
| Transformer (Llama) | 44.08 | 39.98 | 721.1 | 414.7 | Accurate recall while inefficient |
| State Space Models (Mamba) | 42.98 | 19.23 | 4720.8 | 1.9 | Efficient while inaccurate recall |
| A. + Attention heads (sequential) | 44.07 | 45.16 | 776.3 | 156.3 | Enhance recall capabilities |
| B. + Multi-head structure (parallel) | 45.19 | 49.90 | 876.7 | 148.2 | Better balance of two modules |
| C. + Local / global attention | 44.56 | 48.79 | 2399.7 | 41.2 | Boost compute/cache efficiency |
| D. + KV cache sharing | 45.16 | 48.04 | 2756.5 | 39.4 | Cache efficiency |
| E. + Meta tokens | 45.59 | 51.79 | 2695.8 | 40.0 | Learned memory initialization |
| **Scaling to 1.5B model size and 1.5T training tokens** | | | | | |
| F. + Size / data | 60.56 | 64.15 | 664.1 | 78.6 | Further boost task performance |
| G. + Extended context length (2K→8K) | 60.64 | 68.79 | 664.1 | 78.6 | Improve multi-shot and recall tasks |

Table 1: Design roadmap of our Hymba model. We evaluate the models' (1) commonsense reasoning accuracy, averaged over 8 tasks, and (2) recall accuracy, averaged over 2 tasks. The throughput is on NVIDIA A100, sequence length 8k, batch size 128. The cache size is measured with a 8k sequence length, assuming the FP16 format.

Figure 2: (a) Visualize the hybrid-head module in Hymba; (b) Interpret from the memory aspect.

**Efficiency and KV cache optimization.** While attention heads improve task performance, they increase KV cache requirements and reduce throughput. To mitigate this, we optimize the hybrid-head module by combining local and global attention and employing cross-layer KV cache sharing (see Tab. 1 (C)-(D) and Sec. 2.2). This improves throughput by $3\times$ and reduces cache by almost $4\times$.

**Meta tokens.** A set of 128 learnable embeddings is prepended to input tokens, serving as a learned cache initialization to enhance focus on relevant information. These tokens fulfill a dual purpose: (i) they mitigate attention drain by acting as backstop tokens, effectively redistributing attention, and (ii) they encapsulate compressed world knowledge (see Tab. 1 (E) and Sec. 2.3).

**Scaling up model size and data.** Ablation studies were conducted on a 300M-parameter model using 100B training tokens. The final models were trained with 1.5T tokens and scaled up to 1.5B-parameter models (see Tab. 1 (F) and Sec. 2.4).

## 2.1 A FUSED HYBRID-HEAD MODULE

SSM models are efficient but suffer from limited recall capabilities and task performance (Waleffe et al., 2024; Jelassi et al., 2024; Arora et al., 2024a; Ben-Kish et al., 2024) as seen in Tab. 1. Given the high recall resolution of attention, in this step we aim to (1) combine the processing efficiency and context summarization capabilities of SSMs with the high recall resolution of attention, and (2) develop a fused building block to achieve this goal, so it can serve as a fundamental component for constructing future foundation models.

Previous hybrid models (Ren et al., 2024; Glorioso et al., 2024; Lieber et al., 2024) often combine attention and SSMs in a sequential manner. This strategy may lead to information bottlenecks when a layer type that is poorly suited for a specific task cannot effectively process the information. Motivated by the multi-head attention structure in the vanilla Transformer (Vaswani, 2017), where different heads undertake different roles and focus on different contexts (Lv et al., 2024; Merullo et al., 2024), we propose an alternative approach: *fusing attention and SSMs in parallel into a hybrid-head module*, as shown in Fig. 2 (a). The advantage of this design is that different attention and SSM heads can store, retrieve, and process the same piece of information in distinct ways, thereby inheriting the strengths of both operators.

**Design formulation.** We show that the hybrid-head module can be represented by a unified and symmetric formulation. As shown in Fig. 2 (a), given the input sequence $\tilde{X}$, which is the original input sequence $X$ prepended with meta tokens introduced in Sec. 2.3, the input projection $W_{\text{in\_proj}} = [W^Q, W^K, W^V, W^{SSM}, W^G]$ projects $\tilde{X}$ to the query, key, and value of the attention heads using $W^Q, W^K$, and $W^V$, respectively, as well as the input features and gates of the SSM heads using $W^{SSM}$ and $W^G$, respectively.

Following (Vaswani, 2017), the output of attention heads $Y_{\text{attn}}$ can be formulated as:

$$Y_{\text{attn}} = \text{softmax}(QK^T/\sqrt{d}) \, W^V \tilde{X} = M_{\text{attn}} \tilde{X} \qquad (1)$$

where $M_{\text{attn}} = \text{softmax}(QK^T/\sqrt{d}) \, W^V$ and $Q = W^Q \tilde{X}$, $K = W^K \tilde{X}$.

Similar to the attention heads, the SSM heads in our model, for which we adopt Mamba (Gu & Dao, 2023), can also be represented using a data-controlled linear operator $M_{\text{ssm}}$, following (Ali et al.,

2024; Ben-Kish et al., 2024). Specifically, the SSM head output $Y_{\text{ssm}}$ can be formulated as:

$$\alpha^{i,j} = C_i \left( \prod_{k=j+1}^{i} \exp(A\Delta_k) \right) B_j \Delta_j,$$

$$Y_{\text{ssm}} = G \odot \alpha(A, B, C, \Delta) \, W^{SSM} \tilde{X} = M_{\text{ssm}} \tilde{X}, \tag{2}$$

where $M_{\text{ssm}} = G \odot \alpha(A, B, C, \Delta) \, W^{SSM}$, $G = W^G \tilde{X}$ is an output gate, and $A, B, C, \Delta$ are the SSM parameters following the definition in (Gu & Dao, 2023). More specifically, $A$ is a learnable matrix, $B = W_B X_{ssm}$, $C = W_C X_{ssm}$, and $\Delta = \text{Softplus}(W_\Delta X_{ssm})$ with $X_{ssm} = W^{SSM} \tilde{X}$.

We observed that the output magnitudes of the SSM heads, $Y_{\text{ssm}}$, are consistently larger than those of the attention heads, $Y_{\text{attn}}$, as visualized in Fig. 9 in Append. D. To ensure effective fusion, we normalize and re-scale them using learnable vectors to improve training stability, and then average the outputs, followed by a final output projection. The overall formulation of our fused module can be represented symmetrically:

$$Y = W_{\text{out\_proj}} \left( \beta_1 \text{norm}(M_{\text{attn}} \tilde{X}) + \beta_2 \text{norm}(M_{\text{ssm}} \tilde{X}) \right) \tag{3}$$

where $\beta_1$ and $\beta_2$ are learnable vectors that re-scale each channel of the outputs from the attention and SSM heads, respectively. We further explore the optimal ratio of SSMs and attention in hybrid heads in Append. D, and analyze the relative importance of heads in Append. E.

**Interpretation from the memory aspect.** The components in the hybrid-head module can be interpreted as analogous to human brain functions. Specifically, as shown in Fig. 2 (b), the attention heads provide high recall resolution and thus act like snapshot memories in the human brain, storing detailed recollections of a moment or event. In contrast, the SSM heads summarize the context through a constant cache and thus function as fading memories, which gradually forget the details of past events while retaining their core or gist. As shown in Tab. 11 in Append. D, in our Hymba, the summarized global context from fading memories enables allocating more snapshot memories for memorizing local information while maintaining recall capabilities. This is achieved by replacing most global attention with local attention, thus improving memory efficiency.

## 2.2 KV Cache Optimization

Our hybrid-head module improves recall and reasoning capabilities but can compromise memory and throughput efficiency due to the KV cache required by the attention heads. To address this, we aim to reduce the KV cache while maintaining comparable task performance.

**Combine global and local attention.** Local attention, also known as sliding window attention (SWA) (Beltagy et al., 2020), offers a more efficient alternative to global full attention, though it risks losing global context. However, with the presence of SSM heads in our hybrid-head module, which already summarize global context, we can more aggressively replace global full attention with local attention, achieving a better balance between efficiency and performance.

**Exploring the ratio of local attention and global attention.** As shown in Tab. 11 in Append. D, we initially replace global attention in all layers with SWA, which results in a significant degradation in recall capabilities, with accuracy dropping by over 20% on recall-intensive tasks. In response, we progressively reinstate global attention in some layers. Interestingly, as shown in Tab. 1 (C), we find that using global attention in just three layers (i.e., the first, middle, and last layers) is sufficient to recover recall accuracy while maintaining comparable commonsense reasoning accuracy. In turn, this strategy achieves $2.7\times$ throughput and $3.8\times$ cache reduction.

**Cross-layer KV sharing.** Recent works (Liu et al., 2024a) observe that KV cache shares a high similarity between adjacent layers, suggesting that using separate KV caches for each layer leads to both cache and parameter redundancy. In light of this, we employ cross-layer KV sharing (Brandon et al., 2024), where keys and values are shared between consecutive layers (e.g., every two layers share the same KV cache). This strategy reduces both KV memory usage and model parameters, allowing the saved parameters to be reallocated to other model components. As shown in Tab. 1 (D), cross-layer KV sharing improves throughput by $1.15\times$ while maintaining comparable recall accuracy and boosting commonsense accuracy by +0.60%.

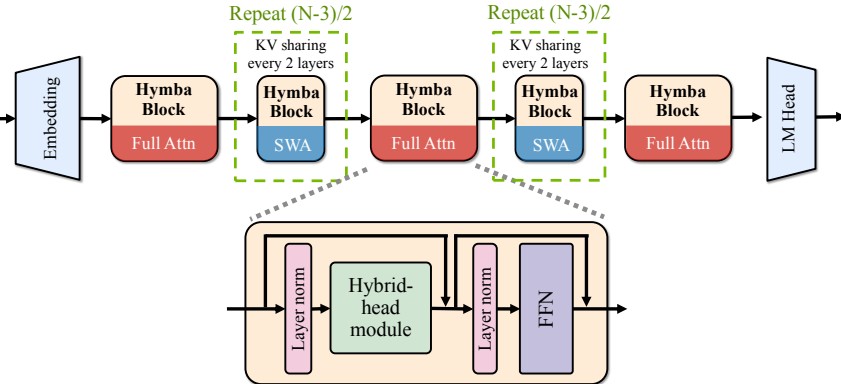

Figure 3: The overall architecture and building block of our Hymba model.

After the above optimization, Hymba's overall architecture is visualized in Fig. 3. It is worth noting that the proposed KV cache optimization strategies are related to the design of the hybrid-head modules. For example, if we apply the same KV cache sharing strategy to the Llama3 model, its average commonsense reasoning accuracy drops from 44.08% to 43.61% and recall accuracy drops from 39.98% to 28.18%. This suggests that, with the help of Mamba heads, attention heads in our hybrid-head modules are more tolerant to those lossy KV cache optimization strategies.

## 2.3 META TOKENS

We observed that the initial tokens, though not semantically important, often receive significant attention scores from subsequent tokens, similar to observations in prior work (Xiao et al., 2023; Han et al., 2024). As shown in Fig.11, more than 50% of the attention is focused on the BOS token for Llama3.2-3B. To address this, we aim to guide the attention to focus more on tokens that meaningfully contribute to task performance. Specifically, we introduce a set of learnable meta tokens $R = [r_1, r_2, \ldots, r_m]$ to serve as the initial tokens. Given the input sequence $X = [x_1, x_2, \ldots, x_n]$, these meta tokens are prepended to the input sequence, forming the modified input sequence:

$$\tilde{X} = [R, X] = [r_1, r_2, \ldots, r_m, x_1, x_2, \ldots, x_n] \tag{4}$$

where $\tilde{X}$ represents the new input sequence for our model. At inference time, since the meta tokens are fixed and appear at the beginning of any input sequences, their computation can be performed offline. Thus, the role of meta tokens at inference can also be viewed as *learned cache initialization* to modulate the subsequent tokens, allowing subsequent tokens to focus more on those that contribute meaningfully to task performance. Similar to the analogy in Sec. 2.1, the meta tokens participate in the attention and SSM calculations of all subsequent tokens, analogous to metamemory in the human brain, which helps recognize where to locate needed information in other memories. We provide further analysis from the memory perspective in Append. G.

**The role of Meta Tokens.** We hypothesize that meta tokens perform the following functions. *Prevent token overwriting.* As shown in (Darcet et al., 2023), attention tends to overwrite and excessively focus on certain tokens, functioning as a garbage collector. This phenomenon was later observed in LLMs and termed "attention sinks" (Xiao et al., 2023; Han et al., 2024). Introducing learnable tokens independent of the input improves the learning of a generalizable garbage collector.

*Exit tokens* to deal with "forced-to-attend". Prepending tokens to the input affects the shape of the softmax function by modifying the denominator. Quiet Attention (Miller, 2023) alters the softmax denominator by adding one, enabling the attention to output zeros. Adding one is equivalent to prepending an all-zero token to the keys and values. Our meta tokens extend this idea by being learnable, allowing the model to optimize the softmax shape.

*Initialization* for KV cache and SSM state. Learning initial tokens can be seen as a form of learned prompt tuning (Lester et al., 2021; Gu et al., 2021c) or learned initialization. For inference, meta tokens are fixed, and the keys and values can be precomputed offline and stored. Task-specific meta tokens can be used, though in this work we use one set for all tasks.

**Meta tokens boost recall capabilities and commonsense reasoning accuracy.** To analyze the impact of meta tokens on the attention mechanism, we visualize the entropy of the attention map

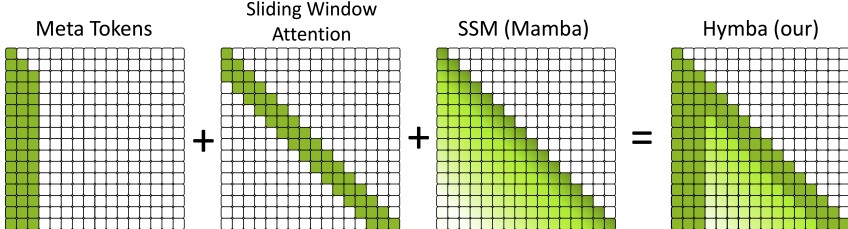

Figure 4: Visualize Hymba's attention map as the contributions of meta tokens, SWA, and Mamba.

for both the attention and SSM heads (Ali et al., 2024; Ben-Kish et al., 2024) before and after introducing meta tokens. Specifically, the attention map entropy reflects the distribution of attention scores across tokens, where lower entropy indicates stronger retrieval effects (Ren et al., 2024), as the attention scores are concentrated around a smaller subset of tokens, and vice versa.

We provide the visualization in Fig. 13 in Append. G, where we observe that, after introducing meta tokens, both the attention and SSM heads exhibit an overall reduction in entropy. Combined with the improved reasoning and recall capabilities shown in Tab. 1 (E), this suggests that meta tokens may help both the attention and SSM heads focus more on a subset of important tokens that contribute most to task performance.

**Hymba attention map.** Hymba's attention pattern can be viewed as a combination of individual components from SWA, meta tokens, and SSM, as shown in Fig. 4. More analysis of Hymba's real attention map and a comparison with the Llama and Jamba models are provided in Append. F. We observe that in vanilla Transformers, attention scores are more concentrated on the beginning-of-sequence token, which is consistent with the findings in (Xiao et al., 2023), and have a higher proportion of local attention scores focusing on the token itself. In Hymba, meta tokens, attention heads, and SSM heads complement each other, leading to a more balanced distribution of attention scores across different types of tokens.

## 2.4 HYMBA MODEL FAMILY

Building on the design insights explored above, we scale up the model sizes and training tokens to deliver the Hymba model family, which includes a 125M model, a 350M model, and a 1.5B model.

We train Hymba-125M/350M/1.5B models using a mix of DCLM-Baseline-1.0 (Li et al., 2024), SmolLM-Corpus (Ben Allal et al., 2024), and a proprietary high-quality dataset, with 1T, 250B, and 50B tokens, respectively. We combine the Warmup-Stable-Decay (WSD) learning rate scheduler (Hu et al., 2024), with maximum and minimum learning rates of 3e-3 and 1e-5, and the data annealing technique (Dubey et al., 2024; Shen et al., 2024) to ensure stable pretraining. We use a sequence length of 2k and a batch size of 2M tokens throughout the training process until the last 100B tokens, where we increase the sequence length to 8k and change the ROPE base following (bloc97, 2023). The overall training pipeline is illustrated in Fig. 5. More details are provided in Append. H.

## 3 EXPERIMENTAL RESULTS

### 3.1 EXPERIMENT SETTINGS

**Baselines.** Our baselines include popular (small) LMs with quadratic attention (e.g., Llama 3.2 (AI, 2024c), SmolLM (Allal et al., 2024b), SmolLM2 (Allal et al., 2024a), AMD-OLMo (Liu et al., 2024b), StableLM (Bellagente et al., 2024), Olmo (Groeneveld et al., 2024), Cosmo (Huggingface,

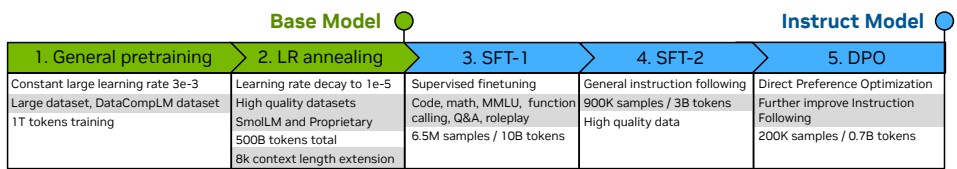

Figure 5: Training pipeline adapted for Hymba family. See Fig. 14 for detailed loss curve.

Table 2: Benchmark Hymba with SOTA small LMs. All models have fewer than 2B parameters, except for Llama-3.2-3B, which is marked as gray. All results are obtained through LM-EVALUATION-HARNESS (Gao et al., 2023). SQuAD-C (SQuAD-Completion) indicates a variant of the SQuAD question answering task proposed by Arora et al. (2024b). The throughput is measured with a 8k sequence length and a 128 batch size on an NVIDIA A100 GPU. The best results are highlighted in **bold**, and the second-best results are highlighted in underline, where Llama-3.2-3B is not included in the ranking due to its 3B model size.

| Model | #Params. | Train tokens | Token/s | Cache (MB) | MMLU 5-shot | ARC-E 0-shot | ARC-C 0-shot | PIQA 0-shot | Wino. 0-shot | Hella. 0-shot | SQuAD-C 1-shot | Avg. |
|---|---|---|---|---|---|---|---|---|---|---|---|---|
| OpenELM-1 | 1.1B | 1.5T | 246 | 346 | 27.06 | 62.37 | 19.54 | 74.76 | 61.80 | 48.37 | 45.38 | 48.47 |
| Rene-v0.1 | 1.3B | 1.5T | 800 | 113 | 32.94 | 67.05 | 31.06 | 76.49 | 62.75 | 51.16 | 48.36 | 52.83 |
| Phi-1.5 | 1.3B | 0.15T | 241 | 1573 | 42.56 | 76.18 | 44.71 | **76.56** | 72.85 | 48.00 | 30.09 | 55.85 |
| SmolLM | 1.7B | 1T | 238 | 1573 | 27.06 | 76.47 | 43.43 | 75.79 | 60.93 | 49.58 | 45.81 | 54.15 |
| Cosmo | 1.8B | 0.2T | 244 | 1573 | 26.10 | 62.42 | 32.94 | 71.76 | 55.80 | 42.90 | 38.51 | 47.20 |
| h2o-danube2 | 1.8B | 2T | 271 | 492 | 40.05 | 70.66 | 33.19 | 76.01 | 66.93 | **53.70** | 49.03 | 55.65 |
| Llama-3.2-1B | 1.2B | 9T | 535 | 262 | 32.12 | 65.53 | 31.39 | 74.43 | 60.69 | 47.72 | 40.18 | 50.29 |
| Qwen2.5 | 1.5B | 18T | 469 | 229 | **60.92** | 75.51 | 41.21 | 75.79 | 63.38 | 50.20 | 49.53 | 59.51 |
| AMD-OLMo | 1.2B | 1.3T | 387 | 1049 | 26.93 | 65.91 | 31.57 | 74.92 | 61.64 | 47.30 | 33.71 | 48.85 |
| SmolLM2 | 1.7B | 11T | 238 | 1573 | 50.29 | **77.78** | 44.71 | 77.09 | 66.38 | 53.55 | 50.50 | 60.04 |
| Llama-3.2-3B | 3.0B | 9T | 191 | 918 | 56.03 | 74.54 | 42.32 | 76.66 | 69.85 | 55.29 | 43.46 | 59.74 |
| **Hymba** | 1.5B | 1.5T | 664 | 79 | 51.19 | 76.94 | **45.90** | **77.31** | 66.61 | 53.55 | **55.93** | **61.06** |

2024), Phi-1.5 (Li et al., 2023), H2O-Danube (Singer et al., 2024), OpenELM (Mehta et al., 2024), and MiniCPM (Hu et al., 2024)), as well as hybrid models (e.g., Rene (AI, 2024a)).

**Benchmark settings.** We adopt two benchmarking settings: (1) In Sec. 3.2, we directly benchmark our delivered Hymba against SOTA public small LMs, and (2) in Sec. 3.3, we train different architectures from scratch with the same dataset, number of layers, model size, and training recipes.

**Benchmark tasks.** In addition to evaluating commonsense reasoning and recall-intensive tasks on our base models, we also evaluate our instruction-tuned models on downstream tasks such as math, function calling, and role-playing in Sec. 3.4.

## 3.2 BENCHMARK WITH SOTA SMALL LMS

We present the benchmark results of our Hymba models with parameter sizes of 125M, 350M, and 1.5B, compared to SOTA small language models within the same size range.

As highlighted in Tab. 2, with only 1.5T pretraining tokens, our Hymba-1.5B model achieves the best performance among all sub-2B LMs and demonstrates better throughput and cache efficiency compared to all transformer-based LMs, with this speedup becoming even more pronounced as the sequence length increases. For instance, compared to the strongest sub-2B baseline, SmolLM2-1.7B, trained on 11T tokens, our Hymba-1.5B, trained on only 1.5T tokens, achieves a 1.02% average accuracy improvement, a 19.91× cache size reduction, and 2.79× throughput. When comparing with small LMs trained on no more than 2T tokens, our model achieves a 5.21%/5.41% average accuracy improvement over the most competitive baselines, Phi-1.5 and h2o-danube2-1.8B, respectively. Additionally, our model even outperforms Llama-3.2-3B, with 1.32% higher average accuracy, an 11.67× cache size reduction, and 3.49× throughput.

We visualize the trade-offs between commonsense reasoning accuracy and cache size/throughput in Fig. 6. In addition, our delivered tiny LMs, Hymba-125M/350M, consistently outperform all LMs of comparable model size, as summarized in Tab. 5 and Tab. 6 in Append. A.1. We have also provided a Hymba-1.5B model trained exclusively on public data in Append. A.2.

## 3.3 BENCHMARK DIFFERENT ARCHITECTURES UNDER THE SAME SETTING

**General and recall-intensive tasks performance comparison.** We do a comprehensive comparison between Hymba and other model architectures, including standard Transformer (Llama3 (AI, 2024b)), pure Mamba (Gu & Dao, 2023; Dao & Gu, 2024), Mamba with FFN, and hybrid architecture with sequential layer stacking (Samba (Ren et al., 2024)) on several downstream tasks. All

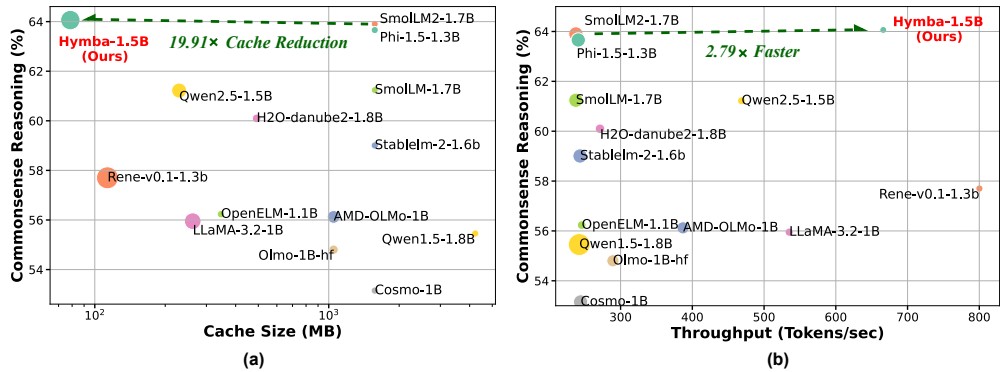

Figure 6: Visualize the trade-off between (a) commonsense reasoning accuracy (avr. ARC-C, ARC-E, PIQA, Hellaswag, OBQA, and Winogrande using (Gao et al., 2023)) and cache size, with throughput represented by the point size of different models, and (b) commonsense reasoning accuracy and throughput, with cache size represented by the point size. The throughput is measured with a 8k sequence length and a 128 batch size on an NVIDIA A100 GPU. The cache size is measured with a 8k sequence length, assuming the FP16 format.

| Task Type | Arch. Style (1B) | Mamba2 | Mamba2 w/ FFN | Llama3 | Samba | Hymba |
|---|---|---|---|---|---|---|
| Language | Wiki. ppl. ↓ | 19.17 | 20.42 | 19.28 | 19.91 | **18.62** |
| | LMB. ppl. ↓ | 12.59 | 14.43 | 13.09 | 12.65 | **10.38** |
| Recall Intensive | SWDE ↑ | 50.24 | 26.43 | **75.95** | 30.00 | 54.29 |
| | SQuAD-C ↑ | 36.43 | 31.40 | 18.70 | 42.33 | **44.71** |
| | Avg. ↑ | 43.34 | 28.92 | 47.33 | 36.17 | **49.50** |
| Common-sense Reasoning and Question-answering | Lambda ↑ | 47.51 | 44.54 | 47.95 | 49.08 | **52.84** |
| | PIQA ↑ | 73.94 | 73.07 | 73.45 | 73.23 | **74.97** |
| | ARC-C ↑ | 38.91 | 37.03 | 39.68 | 39.59 | **41.72** |
| | ARC-E ↑ | 70.96 | 71.00 | 73.74 | 73.36 | **74.12** |
| | Hella. ↑ | 57.73 | 55.83 | 57.64 | 58.49 | **60.05** |
| | Wino. ↑ | **58.48** | 55.56 | 56.20 | 57.54 | 57.85 |
| | TruthfulQA ↑ | 30.75 | 29.86 | 31.64 | 28.84 | **31.76** |
| | SIQA ↑ | 41.86 | 42.22 | 42.22 | 42.48 | **43.24** |
| | Avg. ↑ | 52.52 | 51.14 | 52.82 | 52.83 | **54.57** |

Table 3: Apple-to-apple comparison of our Hymba, pure Mamba2 (Dao & Gu, 2024), Mamba2 with FFN, Llama3 (Dubey et al., 2024) style, and Samba-style (Mamba-FFN-Attn-FFN) (Ren et al., 2024) architectures. All models have 1B parameters and are trained from scratch for 100B tokens from SmolLM-Corpus (Ben Allal et al., 2024) with exactly the same training recipe. All results are obtained through LM-EVALUATION-HARNESS (Gao et al., 2023) using a zero-shot setting. The best and second best results are highlighted in **bold** and underline, respectively.

models have the same number of layers and total parameters to facilitate fair comparison. Models are trained on the same data with the same hyperparameters and under the same codebase. To ensure our conclusions are generally valid, we run comparison experiments at different scales (1B and 300M) and different training datasets (SmolLM-corpus (Ben Allal et al., 2024) and FineWeb (Penedo et al., 2024)) in Tab. 3 and Tab. 8, respectively. We evaluate the models on language modeling, real-world recall-intensive, commonsense reasoning, and question-answering tasks.

As shown in Tab. 3, our Hymba model consistently outperforms other 1B architectures across most tasks, e.g., achieving an average score 1.45% higher than the second-best model at the 300M scale and 1.74% higher at the 1B scale. The ablation study for the 300M scale is in Append. A.

In addition, considering that Mamba models suffer from limited recall capabilities due to their constant-size cache and recurrent nature (Ben-Kish et al., 2024; Arora et al., 2024a; Jelassi et al.,

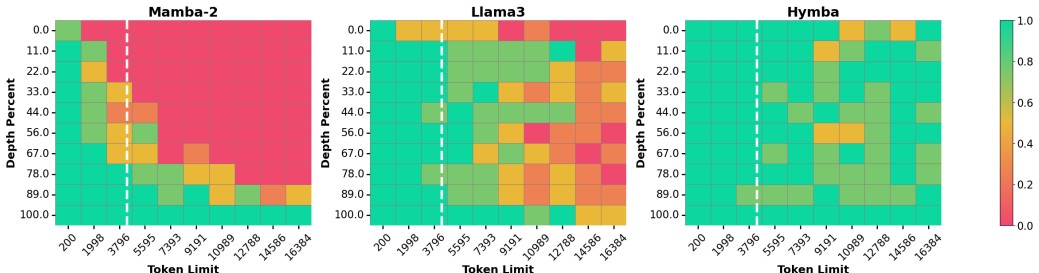

Figure 7: Needle-in-the-haystack performance comparison across different architecture under apple-to-apple setting. The white vertical line represents the finetuning sequence length (4k).

2024), we test the models on two real-world recall-intensive tasks, SWDE (Arora et al., 2024a; Lockard et al., 2019) and SQuAD (Arora et al., 2024a; Rajpurkar et al., 2018), where the former is to to extract semi-structured relations from given raw HTML websites and the latter is to extract answers from a given context passages. Echoing the previous findings, Mamba2 and Mamba2 with FFN architectures under-perform the Transformer model (i.e. Llama3) on these tasks (see Tab. 3). Hymba model augments the Mamba heads with attention heads, which allows the model to have a large effective receptive field to establish long-range dependencies and high-resolution memory to store and retrieve key information in all layers. As a result, Hymba outperforms the Transformer and Samba architectures (where the latter stacks Mamba and attention layers sequentially).

**Needle-in-the-Haystack performance comparison.** We further do an apple-to-apple comparison between Hymba, Mamba2, and Llama3 on the synthetic retrieval task, needle-in-the-haystack. A random and informative sentence (i.e., needle) is inserted into a long document (i.e., haystack) and the model is required to retrieve the needle from the haystack to answer the questions. All models are of size 1B and trained with the same setting: (i.) pretrain is done with 1k sequence length; (ii.) finetune with 4k sequence length; (iii.) test with up to 16k sequence length. If model has ROPE, then we adjust the ROPE as in (Liu et al., 2023) during finetuning.

As shown in Fig. 7, the Hymba model significantly outperforms the Mamba2 and Llama3 models. While the Mamba2 model has good extrapolation capabilities when the needle is inserted in the end of the haystack, it struggles to retrieve the needle when the needle is in the beginning or middle of the haystack. In contrast, Llama3 model has limited extrapolation capabilities (Peng et al., 2023b; Liu et al., 2023; Zhang et al., 2024) and struggles to the "lost in the middle" (Liu et al., 2024c) scenario. We provide more real-world long-context tasks evaluation in Append. B, where we show that Hymba has comparable or better length generalization capabilities comparing to vanilla Transformers under similar training length.

### 3.4 BENCHMARK INSTRUCTION-TUNED MODELS

**Implementation details of post-training.** We post-train Hymba-1.5B base model with a two-stage strategy: the first full-finetuning (FFT) stage and another DPO (Rafailov et al., 2024) training. The learning rates are 5e-5 and 3e-6 for FFT and DPO, respectively. To accelerate training, we follow the training recipe (Tunstall et al., 2023; Diao et al., 2024; Dong et al., 2024) to pack the samples and use a block size of 8192. We compare Hymba-1.5B-Instruct with competitive lightweight instruction-tuned models, i.e., Llama-3.2-1B-Instruct (AI, 2024c), OpenELM-1-1B-Instruct (Mehta et al., 2024), Qwen2.5-1.5B-Instruct (Team, 2024), and SmolLM-1.7B-Instruct (Allal et al., 2024b). We test the instruction-tuned models on MMLU (5-shot), IFEval, GSM8K (5-shot), GPQA (0-shot), and Berkeley Function-Calling Leaderboard v2 (BFCLv2) (Yan et al., 2024). More details about the experimental settings, baseline models, and evaluation tasks are shown in Append. H.

**Evaluation results.** The evaluation results are shown in Tab. 4. In general, Hymba-1.5B-Instruct achieves the highest performance on an average of all tasks, outperforming the previous SoTA model, Qwen2.5-Instruct, by around 2%. It demonstrates a great ability in math, reasoning, and function calling, with the best-in-class performance.

In addition to full finetuning, we evaluate whether Hymba is compatible with parameter-efficient finetuning methods by finetuning the post-trained Hymba on RoleBench (Wang et al., 2023) using DoRA (Liu et al., 2024d). Results are provided in Append. C, where we find that DoRA-finetuned Hymba significantly outperforms larger models.

Table 4: The comparison between lightweight instruction-tuned models. The best and second-best results are highlighted in **bold** and underline, respectively. *OpenELM and SmolLM cannot understand function calling, leading to zero accuracy in most categories.

| Model | #Params | MMLU ↑ | IFEval ↑ | GSM8K ↑ | GPQA ↑ | BFCLv2 ↑ | Avg. ↑ |
|---|---|---|---|---|---|---|---|
| SmolLM | 1.7B | 27.80 | 25.16 | 1.36 | 25.67 | -* | 20.00 |
| OpenELM | 1.1B | 25.65 | 6.25 | 56.03 | 21.62 | -* | 27.39 |
| Llama-3.2 | 1.2B | 44.41 | **58.92** | 42.99 | 24.11 | 20.27 | 38.14 |
| Qwen2.5 | 1.5B | **59.73** | 46.78 | 56.03 | 30.13 | 43.85 | 47.30 |
| SmolLM2 | 1.7B | 49.11 | 55.06 | 47.68 | 29.24 | 22.83 | 40.78 |
| Hymba-1.5B | 1.5B | 52.79 | 57.14 | **58.76** | **31.03** | **46.40** | **49.22** |

## 4 RELATED WORKS

**Large language models.** Prior to the rise of LLMs, transformer-based models (Vaswani, 2017; Devlin et al., 2018; Raffel et al., 2020; Roberts et al., 2022) proved highly effective at capturing relationships between tokens in complex sequences through the use of the attention mechanism (Vaswani, 2017). These models also demonstrated considerable scalability (Qin et al., 2023; Kaplan et al., 2020; Biderman et al., 2023) in terms of both model size and the volume of pretraining data. This scalability paved the way for the development of LLMs, such as GLM (Du et al., 2021), OPT (Zhang et al., 2022), Mistral (Jiang et al., 2023), the Llama series (Touvron et al., 2023; AI, 2024b), Gemma (Team et al., 2024), and GPT-4 (Achiam et al., 2023), which showcase remarkable zero-shot and few-shot in-context learning abilities.

**Efficient language model architectures.** The quadratic computational complexity and the linearly increasing KV cache size of attention modules with longer sequences limit their processing efficiency. To address this, efficient LMs featuring sub-quadratic complexity in sequence length and strong scaling properties have emerged (Peng et al., 2023a; Sun et al., 2023; Gu & Dao, 2023; Dao & Gu, 2024; Yang et al., 2023; Katharopoulos et al., 2020). As pointed out by (Gu & Dao, 2023), popular efficient LM architectures such as RWKV (Peng et al., 2023a) and RetNet (Sun et al., 2023) can be viewed as variants of SSMs (Gu et al., 2021a;b). Mamba (Gu & Dao, 2023), one of the most widely used SSMs, improves upon previous SSMs by selectively propagating or forgetting information along the sequence length in an input-dependent manner. Follow-up works such as Mamba2 (Dao & Gu, 2024) and GLA (Yang et al., 2023) introduce more hardware-friendly gating mechanisms to enhance training throughput over Mamba.

**Hybrid language models.** To combine the processing efficiency of SSMs with the recall capabilities of transformers, an emerging trend is the creation of hybrid models that incorporate both types of operators. Specifically, (Park et al., 2024; Waleffe et al., 2024) propose hybrid models that interleave Mamba and attention modules to improve commonsense reasoning and in-context learning capabilities. Jamba (Lieber et al., 2024) and Zamba (Glorioso et al., 2024) develop sequentially stacked Mamba-Attention hybrid models. Samba (Ren et al., 2024) introduces a structure that sequentially stacks Mamba, SWA, and MLP layers by repeating the Mamba-MLP-SWA-MLP structure, achieving constant throughput as sequence lengths increase. Other recent work has also explored hybrid models that mix either linear RNNs or convolutions with attention (De et al., 2024; Pilault et al., 2024; Saon et al., 2023; Yang et al., 2024).

## 5 CONCLUSION

In this work, we present Hymba, a new family of small LMs featuring a hybrid-head architecture that combines the high-resolution recall capabilities of attention heads with the efficient context summarization of SSM heads. To further optimize the performance of Hymba, we introduce learnable meta tokens to enhance the model's focus on salient information and propose a series of KV cache optimization techniques. Through the roadmap of Hymba, comprehensive evaluations, and ablation studies, we demonstrate that Hymba sets new SOTA performance across a wide range of tasks, achieving superior results in both accuracy and efficiency. Additionally, our work provides detailed insights into the advantages of hybrid-head architectures, offering a promising direction for future research in efficient language and multi-modal models.

## 6 ACKNOWLEDGMENTS

This work would not have been possible without additional contributions from many people at NVIDIA, including Hanah Zhang, Maksim Khadkevich, Mohammad Shoeybi, Mostofa Patwary, Nikolaus Binder, Chenhan Yu, Meredith Price, and Oluwatobi Olabiyi.

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

# A    EXTENSIVE BENCHMARK FOR MORE HYMBA MODEL VARIANTS

## A.1    COMPARISON WITH SOTA TINY LMS AT 350M AND 125M SCALES

Besides our 1.5B model, we also evaluate the 350M and 125M Hymba models on a diverse set of benchmarks in Tab. 5 and Tab. 6, respectively. Consistent with the results of our 1.5B model, Hymba-350M/125M models outperform the SOTA tiny LMs across most of tasks and achieve the best average score. This indicates that our Hymba scales effectively across different model sizes.

Table 5: Benchmark Hymba with SOTA tiny LMs, all of which have fewer than 200M parameters. All results are obtained through HUGGINGFACE/LIGHTEVAL, following (Allal et al., 2024b).

| Model | #Params. | MMLU (cloze) ↑ | ARC (c+e) ↑ | PIQA ↑ | Hella. ↑ | OBQA ↑ | Wino. ↑ | Avg. ↑ |
|---|---|---|---|---|---|---|---|---|
| Mamba-130m-hf | 130M | 27.41 | 33.01 | 63.33 | 33.86 | 30.40 | 51.54 | 42.43 |
| Cerebras-GPT | 111M | 25.56 | 27.75 | 58.16 | 26.32 | 25.40 | 50.28 | 37.58 |
| GPT-neo | 125M | 27.25 | 31.30 | 62.35 | 29.68 | 29.20 | 51.54 | 40.81 |
| LaMini-GPT | 124M | 26.47 | 33.26 | 62.89 | 30.05 | 27.80 | 50.75 | 40.95 |
| Opt | 125M | 25.67 | 31.25 | 61.97 | 31.04 | 29.00 | 53.20 | 41.29 |
| GPT2 | 137M | 26.29 | 31.09 | 62.51 | 29.76 | 29.40 | 49.72 | 40.50 |
| Pythia | 160M | 26.68 | 31.92 | 61.64 | 29.55 | 27.80 | 49.49 | 40.08 |
| MobileLM | 125M | - | 35.51 | 65.30 | 38.90 | **39.50** | **53.10** | 46.46 |
| SmolLM | 135M | 30.23 | 43.99 | **69.60** | 42.30 | 33.60 | 52.70 | 48.44 |
| Hymba | 125M | **31.12** | **44.95** | 68.50 | **45.54** | 35.52 | 52.25 | **49.35** |

Table 6: Benchmark Hymba with SOTA tiny LMs, all of which have fewer than 400M parameters. All results are obtained through HUGGINGFACE/LIGHTEVAL, following (Allal et al., 2024b).

| Model | #Params. | MMLU (cloze) ↑ | ARC (c+e) ↑ | PIQA ↑ | Hella. ↑ | OBQA ↑ | Wino. ↑ | Avg. ↑ |
|---|---|---|---|---|---|---|---|---|
| Bloom | 560M | 27.49 | 32.86 | 65.13 | 35.98 | 28.80 | 51.70 | 42.89 |
| Cerebras-GPT-256M | 256M | 25.91 | 29.69 | 61.37 | 28.44 | 28.00 | 51.62 | 39.82 |
| Cerebras-GPT-590M | 590M | 26.93 | 32.40 | 62.84 | 31.99 | 28.40 | 50.12 | 41.15 |
| Opt | 350M | 26.57 | 31.94 | 64.36 | 36.09 | 27.80 | 52.57 | 42.55 |
| Pythia | 410M | 28.94 | 35.05 | 66.92 | 39.21 | 28.40 | 52.80 | 44.48 |
| GPT2-medium | 380M | 27.77 | 34.30 | 66.38 | 37.06 | 31.20 | 49.49 | 43.69 |
| MobileLM | 350M | - | 43.65 | 68.60 | 49.60 | **40.00** | 57.60 | 51.89 |
| SmolLM | 360M | 34.17 | 51.10 | 72.00 | 53.80 | 37.20 | 53.70 | 53.56 |
| Hymba | 350M | **34.54** | **52.46** | **72.91** | **55.08** | 38.40 | **57.85** | **55.34** |

## A.2    EVALUATING HYMBA-1.5B TRAINED ON PUBLIC DATA ONLY

We have also trained our Hymba-1.5B model exclusively on public data and evaluated its performance. Specifically, following the training settings in Sec. 2.4, we train Hymba-1.5B on DCLM-Baseline-1.0 (Li et al., 2024) for 1T tokens in the first phase and on SmolLM-Corpus (Ben Allal et al., 2024) for 500B tokens in the second phase, keeping all other settings the same. The results are summarized in Tab. 7, where only the most competitive baselines from Tab. 2 are included. We observe that (1) Hymba-1.5B trained exclusively on public data only still surpasses all baseline small LMs in terms of average accuracy; and (2) Hymba-1.5B trained on public data primarily suffers from performance drops on 5-shot MMLU compared to the version trained on all data, including our proprietary dataset. This suggests that the public data used may lack sufficient factual knowledge, which is supplemented by our proprietary one.

Table 7: Benchmark Hymba-1.5B trained with all data and public data only against SOTA small LMs. All models have fewer than 2B parameters, except for Llama-3.2-3B, which is marked in gray. The settings follow Tab. 2 in our main paper and we only include the most competitive baselines here. **Hymba (Public Data)** refers to our model trained exclusively on public datasets, without using our proprietary high-quality dataset.

| Model | #Params. | Train tokens | Token/s | Cache (MB) | MMLU 5-shot | ARC-E 0-shot | ARC-C 0-shot | PIQA 0-shot | Wino. 0-shot | Hella. 0-shot | SQuAD-C 1-shot | Avg. |
|---|---|---|---|---|---|---|---|---|---|---|---|---|
| Phi-1.5 | 1.3B | 0.15T | 241 | 1573 | 42.56 | 76.18 | 44.71 | 76.56 | **72.85** | 48.00 | 30.09 | 55.85 |
| h2o-danube2 | 1.8B | 2T | 271 | 492 | 40.05 | 70.66 | 33.19 | 76.01 | 66.93 | 53.70 | 49.03 | 55.65 |
| Qwen2.5 | 1.5B | 18T | 469 | 229 | **60.92** | 75.51 | 41.21 | 75.79 | 63.38 | 50.20 | 49.53 | 59.51 |
| SmolLM2 | 1.7B | 11T | 238 | 1573 | 50.29 | 77.78 | 44.71 | 77.09 | 66.38 | 53.55 | 50.50 | 60.04 |
| Llama-3.2-3B | 3.0B | 9T | 191 | 918 | 56.03 | 74.54 | 42.32 | 76.66 | 69.85 | 55.29 | 43.46 | 59.74 |
| **Hymba** | 1.5B | 1.5T | 664 | 79 | 51.19 | 76.94 | 45.90 | 77.31 | 66.61 | 53.55 | 55.93 | **61.06** |
| **Hymba (Public Data)** | 1.5B | 1.5T | 664 | 79 | 44.31 | **78.58** | **47.01** | **77.53** | 64.56 | 53.89 | 59.82 | 60.81 |

Table 8: Apple-to-apple comparison of our Hymba, pure Mamba (Gu & Dao, 2023), Mamba with FFN, Llama3 (Dubey et al., 2024) style, and Samba-style (Mamba-FFN-Attn-FFN) (Ren et al., 2024) architectures. All models have 300M parameters and are trained for 100B tokens from FineWeb dataset (Penedo et al., 2024) with exactly the same training recipes. All results are obtained through LM-EVALUATION-HARNESS (Gao et al., 2023). The best and second best results are highlighted in bold and underline, respectively.

| Task Type | Arch. Style (300M) | Mamba | Mamba w/ FFN | Llama3 | Samba | SMA | Hymba |
|---|---|---|---|---|---|---|---|
| Language | Wiki. ppl. ↓ | 30.78 | 33.41 | 30.04 | 31.41 | 29.75 | **28.53** |
| | LMB. ppl. ↓ | 19.95 | 23.64 | 20.53 | 19.75 | 20.85 | **15.45** |
| Recall Intensive | SQuAD-C ↑ | 21.31 | 17.56 | 22.10 | 39.88 | 44.44 | **45.24** |
| | SWDE ↑ | 17.14 | 13.10 | 57.86 | 22.14 | 55.48 | **58.33** |
| | Avg. ↑ | 19.23 | 15.33 | 39.98 | 31.01 | 49.96 | **51.79** |
| Common-sense Reasoning and Question-answering | Lambda ↑ | 38.95 | 36.37 | 40.15 | 40.59 | 40.40 | **44.67** |
| | PIQA ↑ | 69.64 | 69.26 | 70.29 | 69.86 | 69.80 | **70.73** |
| | ARC-C ↑ | 24.91 | 25.00 | 24.83 | 25.76 | 25.96 | **26.28** |
| | ARC-E ↑ | 50.67 | 50.34 | 50.24 | 49.79 | 49.62 | **53.20** |
| | Hella. ↑ | 44.95 | 44.08 | 45.69 | 46.45 | 46.42 | **48.23** |
| | Wino. ↑ | 51.70 | 51.78 | 52.64 | 52.49 | 52.72 | **53.35** |
| | TruthfulQA ↑ | 23.86 | 26.23 | **28.97** | 27.27 | 26.47 | 27.87 |
| | SIQA ↑ | 39.20 | 39.53 | 39.66 | 39.92 | **41.25** | 39.92 |
| | Avg. | 42.98 | 42.82 | 44.08 | 44.02 | 44.08 | **45.53** |

## A.3 APPLE-TO-APPLE COMPARISON WITH OTHER ARCHITECTURES AT 300M SCALE

In addition to the apple-to-apple architecture comparison under the same settings with a 1B model size in Sec. 3.3 of our main paper, we also validate the superiority of our architecture at the 300M size. Specifically, we train different 300M model architectures on 100B tokens from FineWeb (Penedo et al., 2024). We set peak learning rates to 5e-4 and use warmup and cosine decay scheduler. The training sequence length is set to 1K. For models with sliding window attention, we set the sliding window size as 256. Since Samba (Ren et al., 2024) only has local attention, we further build a variant of Samba where we replace its first, last, and middle local attention with global attention to ensure a fair comparison with our model. We call this variant Sequential-Mix-attention (SMA). As shown in Tab. 8, Hymba achieves the best performance in almost all tasks (with a second-best result in one task), yielding an average accuracy boost of +1.45% compared to the strongest baseline.

Table 9: Benchmark Hymba-1.5B and other models on real-world long-context tasks from Long-Bench (Bai et al., 2023).

| Model | GovReport (Rouge-L) | MultiNews (Rouge-L) | QMSum (Rouge-L) | TriviaQA (F1) | SAMSum (Rouge-L) | TREC (Acc) | LSHT (Acc) |
|-------|---------------------|---------------------|-----------------|---------------|------------------|------------|------------|
| SmolLM-1.7B | 4.77 | 12.79 | 8.55 | 1.97 | 3.23 | 1.00 | 0.00 |
| h2o-danube-1.8B | 12.41 | 14.28 | 17.01 | 68.24 | 11.46 | 56.00 | 10.50 |
| Hymba-1.5B | 13.95 | 19.24 | 17.29 | 76.82 | 35.21 | 56.22 | 11.00 |

## B    PERFORMANCE ON REAL-WORLD LONG-CONTEXT TASKS

We evaluate Hymba-1.5B on a broader range of long-context tasks, including summarization and few-shot learning tasks from LongBench (Bai et al., 2023). Specifically, we finetune Hymba-1.5B on an 8k context length using 50B tokens from the SmolLM corpus and benchmark it against the best-performing models: h2o-danube2-1.8B (trained on a 16k context length) and SmolLM-1.7B (trained on a 2k context length). We evaluate all models on three English summarization tasks and four few-shot learning tasks from LongBench.

As shown in Tab. 9, Hymba achieves the best performance across both task types, even surpassing h2o-danube2-1.8B, which has a much larger KV cache size and was trained on a longer context length. Additionally, we note that Hymba's long-context performance can be further improved by finetuning on longer sequences, which will be a focus in future releases.

Table 10: The comparison between DoRA-finetuned Hymba and baselines on RoleBench. All baseline results are from Wang et al. (2023).

| Model | #Params | Instruction Generalization | Role Generalization |
|-------|---------|----------------------------|---------------------|
| Llama-7B | 7B | 19.2 | 19.3 |
| Aplaca-7B | 7B | 25.6 | 24.5 |
| Vicuna-13B | 13B | 25.0 | 24.3 |
| Llama2-7B-chat | 7B | 18.8 | 20.5 |
| RoleLlama-7B | 7B | 35.5 | 33.5 |
| Hymba-DoRA | 1.5B | **40.0** | **37.9** |

## C    DORA-FINETUNING OF HYMBA ON ROLE-PLAY TASKS

We conduct experiments to evaluate whether Hymba is compatible with DoRA (Liu et al., 2024d), a parameter-efficient finetuning method that updates pretrained models using a minimal set of parameters. This approach is especially well-suited for on-device finetuning scenarios where computational resources are constrained. Additionally, DoRA significantly reduces storage requirements for saving multiple downstream models, as it only requires storing the finetuned DoRA parameters, which constitute less than 10% of the original model's total parameters. Specifically, we further finetune the instruction-tuned Hymba on RoleBench (Wang et al., 2023) using DoRA to enhance its role-playing capabilities. The training set of RoleBench is used for training, and the model is evaluated on two sub-tasks: instruction generalization (Inst. Gene.) and role generalization (Role. Gene.).

As shown in the Tab. 10, our Hymba-DoRA significantly outperforms larger models. For instance, DoRA-finetuned Hymba achieves scores of 40.0% / 37.9% on instruction generalization/role generalization, outperforming RoleLlama-7B (Wang et al., 2023) by 4.5%, and 4.4% respectively. This indicates the strong generalization of our model and the effectiveness of using parameter-efficient finetuning techniques to further enhance its performance.

Table 11: Ablation study of the design choices of Hymba. The design finally adopted by Hymba is highlighted in **bold**. Specifically, the task lists are the same as those in Tab. 3. The throughput is measured with a 8k sequence length and a 128 batch size on an NVIDIA A100 GPU. The cache size is measured with a 8k sequence length, assuming the FP16 format.

| Design Factor | | Configuration | Param. Ratio Attn:Mamba | Avg. (General) ↑ | Avg. (Recall) ↑ | Throughput (Token/s) ↑ | Cache (MB) ↓ |
|---|---|---|---|---|---|---|---|
| Attn/Mamba Ratio | 1) | Mamba Heads Only | 0:1 | 42.98 | 19.23 | 4720.8 | 1.87 |
| | 2) | Mamba + 4 Attn Heads | 1:8.48 | 44.20 | 44.65 | 3278.1 | 99.09 |
| | 3) | Mamba + 8 Attn Heads | 1:4.24 | 44.95 | 52.53 | 1816.5 | 197.39 |
| | 4) | Mamba + 16 Attn Heads | 1:2.12 | 45.08 | 56.46 | 656.6 | 394.00 |
| | **5)** | **4) + GQA** | 1:3.64 | 45.19 | 49.90 | 876.7 | 148.24 |
| | 6) | Attn Heads Only (Llama) | 1:0 | 44.08 | 39.98 | 721.1 | 414.72 |
| Sliding Window | 7) | 5) + All SWA's | 1:3.64 | 44.42 | 29.78 | 4485.09 | 5.51 |
| | **8)** | **5) + SWA's + Full Attn** | 1:3.64 | 44.56 | 48.79 | 2399.7 | 41.19 |
| | **9)** | **8) + Cross-layer KV sharing** | 1:5.23 | 45.16 | 48.04 | 2756.5 | 39.42 |
| | 10) | 6) + Same KV compression | 1:0 | 43.60 | 28.18 | 3710.0 | 28.98 |
| Fusion | 11) | 9) Replace Mean by Concat | 1: 5.82 | 44.56 | 48.94 | 1413.9 | 39.42 |
| Meta Tokens | 12) | 1) + Meta Tokens | 0:1 | 44.01 | 19.34 | 4712.8 | 1.87 |
| | **13)** | **9) + Meta Tokens** | 1:5.23 | 45.53 | 51.79 | 2695.8 | 40.01 |

## D    ABLATION STUDIES OF OUR HYMBA ARCHITECTURE

We perform further ablation studies and analyses of the design factors in our Hymba.

**Parallel vs. sequential fusion.** We compare the hybrid-head module with a sequential counterpart, which interleaves local attention and Mamba layers as adopted by (Ren et al., 2024), by calculating the models' effective receptive field (ERF) and their overall cache size. All the compared models have the same parameter size and are training from scratch using exactly the same training recipe. ERF is an empirical measure of the averaged distance among tokens that allows effective information propagation (Ben-Kish et al., 2024; Dosovitskiy, 2020) defined as the following,

$$ERF \approx \sum_{n \leq N} \sum_{h \leq H} \sum_{s \leq S} \frac{2M^h (S, s) \cdot (S - s) \cdot (N - n + 1)}{HN (N + 1)}, \tag{5}$$

where $S$ is index of the last token in the sequence, $N$ is index of the last layer in the model, and $M^h(S, s)$ is the normalized attention score between token $s$ and the last token in head $h$.

As shown in Fig. 8, we observe that (1) in line with common intuitions, Llama3 exhibits a notably larger ERF compared to Mamba due to its higher recall resolution, albeit at the cost of a larger cache size; (2) our hybrid-head structure demonstrates the best ERF across the four designs, with an order of magnitude larger ERF while maintaining a cache size comparable to the sequential structure. This suggests that the parallel structure can better leverage the limited cache size to capture longer and more complex relationships among tokens compared to the sequential one. The differences in ERF are also reflected in task accuracy: According to Tab. 1, the hybrid-head design (Tab. 1 (B)) improves commonsense reasoning and recall accuracy by +1.08% and 4.74%, respectively, over the sequential design (Tab. 1 (A)). Based on this benchmarking and analysis, we adopt the hybrid-head module as our basic building block.

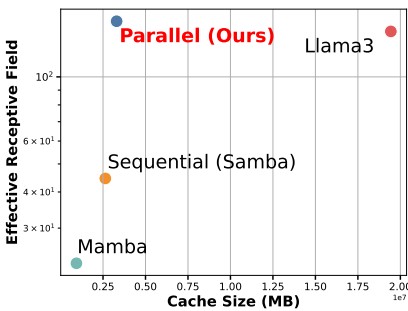

Figure 8: Visualize the ERF and cache size trade-off.

**The ratio of SSMs and attention in hybrid heads.** To determine the proper number of attention heads, we start with a Mamba model and gradually replace Mamba's hidden dimensions with attention heads, maintaining the same overall model size. As shown in Tab. 11 (1)~ (4), we observe that model performance improves as the ratio of attention parameters increases and gradually saturates when the parameter ratio of attention to Mamba reaches 1:2.12. We stop introducing more attention heads, considering that adding more would bring increased memory overhead.

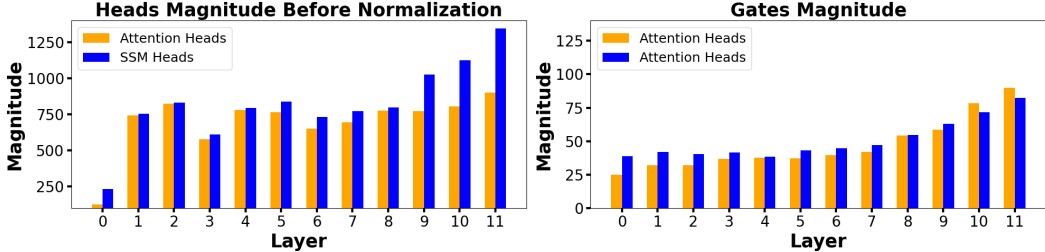

Figure 9: Left: Visualization of output magnitudes of attention and SSM heads. SSM heads consistently have higher output magnitude than attention heads due to their structure. Right: Visualization of attention and SSM heads' gate magnitudes. Through model learning, the relative magnitudes of attention and SSM gates vary across different layers.

There are two interesting observations: (1) Although the attention-only model outperforms the Mamba-only model, the hybrid model with both attention and Mamba heads achieves the best performance; (2) with further KV cache optimization, the ratio of attention heads decreases further. In our final model, attention heads occupy no more than 1/5 of the Mamba heads, yet significantly boost both recall and commonsense reasoning compared to the vanilla Mamba. This suggests that the hybrid model leverages the strengths and diversity of both attention and SSM heads, achieving a better trade-off between efficiency and performance.

**The hybrid-head fusion strategy.** We have explored two straightforward methods to fuse the outputs of attention and SSM heads: concatenation and mean. For concatenation, we combine the outputs of all heads and use a linear layer to project the concatenated output to the final output dimension. However, the parameter size of the linear layer increases with both the number of heads and the head dimensions. Additionally, based on the empirical comparison between Tab. 11 (9) and (11), the performance of concatenation fusion is not better than the simple mean fusion. Therefore, we adopt the mean fusion strategy in our final design.

**Impact of KV cache optimization.** After applying a series of KV cache optimization techniques, moving from Tab. 11 (5) to Tab. 11 (9), we observe that our Hymba maintains comparable recall and commonsense reasoning accuracy while being $2.74\times$ faster. In contrast, applying the same KV cache optimization to a pure Transformer, as seen in the comparison between Tab. 11 (6) and (10), results in a recall accuracy drop of 10% or more and degraded commonsense reasoning accuracy. This supports our analysis in Sec. 2.2, showing that the presence of SSM heads in our hybrid-head module has already summarized the global context, allowing us to more aggressively replace global full attention with local attention in our hybrid model.

# E   HEAD IMPORTANCE ANALYSIS

**Setup.** To understand how hybrid heads contribute to the final task performance, we zero out the attention or SSM heads in each layer by setting $\beta_1$ or $\beta_2$ in Eq. 3 to 0 and record the final accuracy. We consider four datasets, which are presented in Fig. 10, and the task performance is measured using 1000 samples from each task, evaluated with lm-evaluation-harness (Gao et al., 2023) in a zero-shot setting.

**Observations.** As shown in Fig. 10, we observe that (1) the relative importance of attention/SSM heads in the same layer, indicated by the change in task performance before and after being removed, may vary across different tasks. In other words, the relative importance of attention/SSM heads in the same layer is input-adaptive, indicating that different types of heads learn to serve different roles and undertake different responsibilities when handling various inputs; (2) The SSM head in the first layer is critical for language modeling and removing it causes a substantial increase in PPL or a substantial drop in accuracy (to random guess levels). Generally, removing one attention/SSM head results in a 0.46%/1.2% reduction in accuracy averaged across all layers and tasks, respectively.

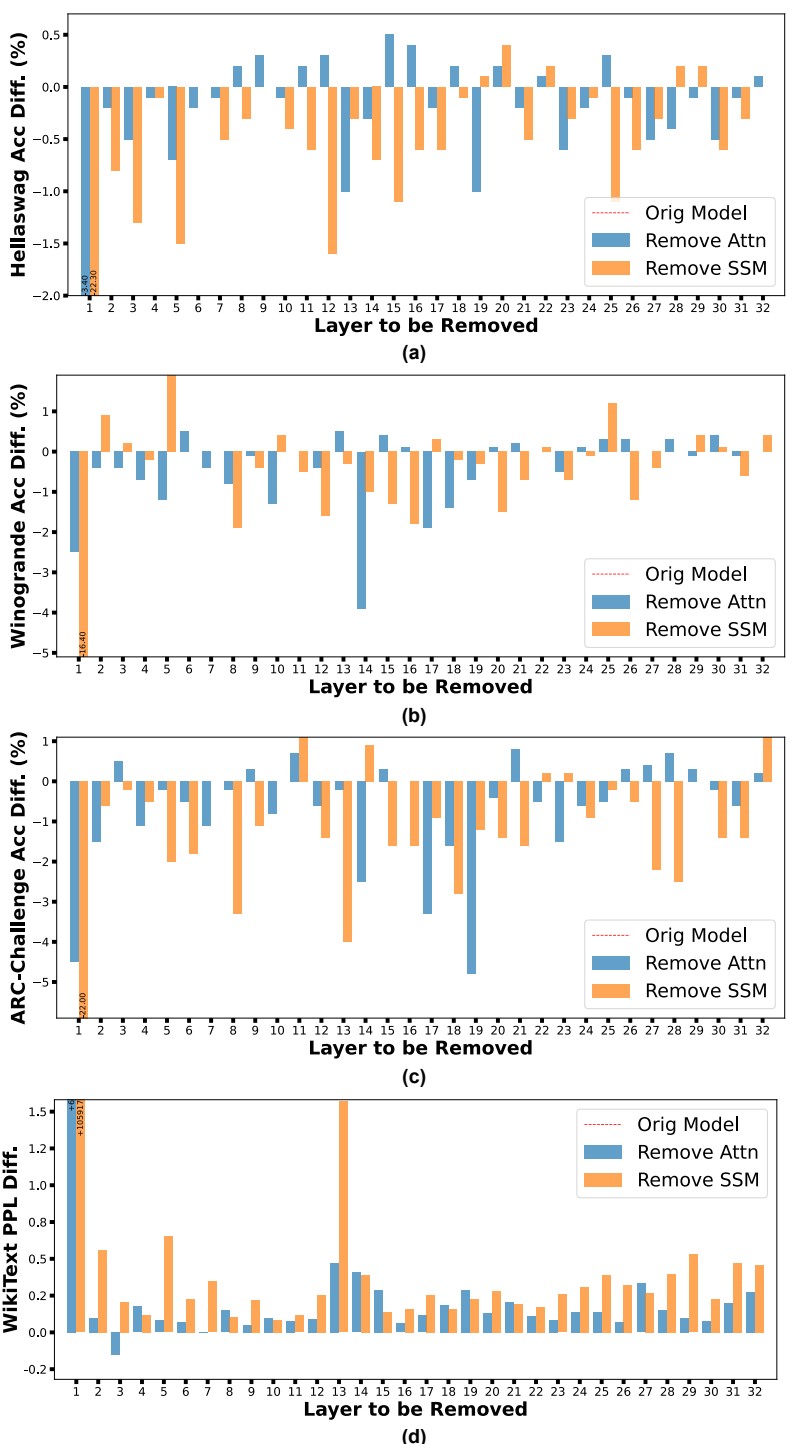

Figure 10: Visualize the task performance difference across three tasks after removing the Attention or SSM heads in each layer. The task performance is measured using 1000 samples from each task. Note that removing critical modules in specific layers causes a significant gap compared to others, making their bars fall outside the box. For such layers, we annotate the task performance with text.

## F  HYMBA ATTENTION MAP VISUALIZATION

In this section, we visualize the actual attention map of Hymba and compare it with those of the Llama and Jamba (Lieber et al., 2024) models. Specifically, we categorize elements in the attention

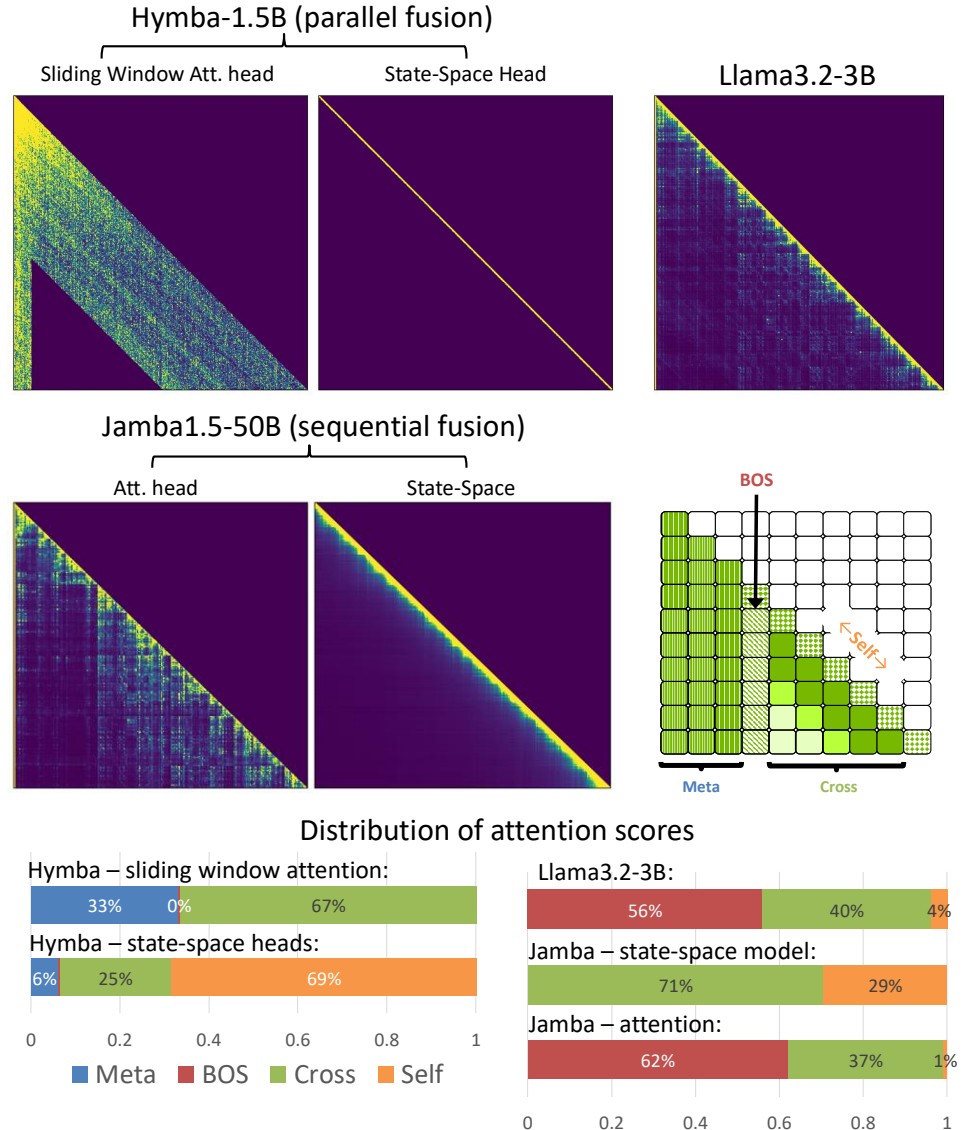

Figure 11: Sum of attention score from different categories (i.e., 'Meta', 'BOS', 'Self', 'Cross') in Llama-3.2-3B, Jamba, and Hymba-1.5B.

map into four types: (1) 'Meta': attention scores from all real tokens to meta tokens. This category reflects the model's preference for attending to meta tokens. In attention map, they are usually located in the first few columns (e.g., 128 for Hymba) if a model has meta tokens. (2) 'BOS': attention scores from all real tokens to the beginning-of-sequence token. In the attention map, they are usually located in the first column right after the meta tokens. (3) 'Self': attention scores from all real tokens to themselves. In the attention map, they are usually located in the diagonal line. (4) 'Cross': attention scores from all real tokens to other real tokens. In the attention map, they are usually located in the off-diagonal area.

In Fig. 11, we visualize the real attention maps from Llama-3.2-3B and Hymba-1.5B on texts from Oliver Twist Chapter 29 (Dickens, 1868) and sum up the attention scores from different categories. The summed scores are normalized by the context length. For SSM heads, we follow (Ben-Kish et al., 2024) and (Zimerman et al., 2024) to calculate their attention maps and normalize the attention maps to ensure each row sums to 1.

We observe that the attention pattern of Hymba is significantly different from the vanilla Transformers. In vanilla Transformers, attention scores are more concentrated on 'BOS', which is consistent

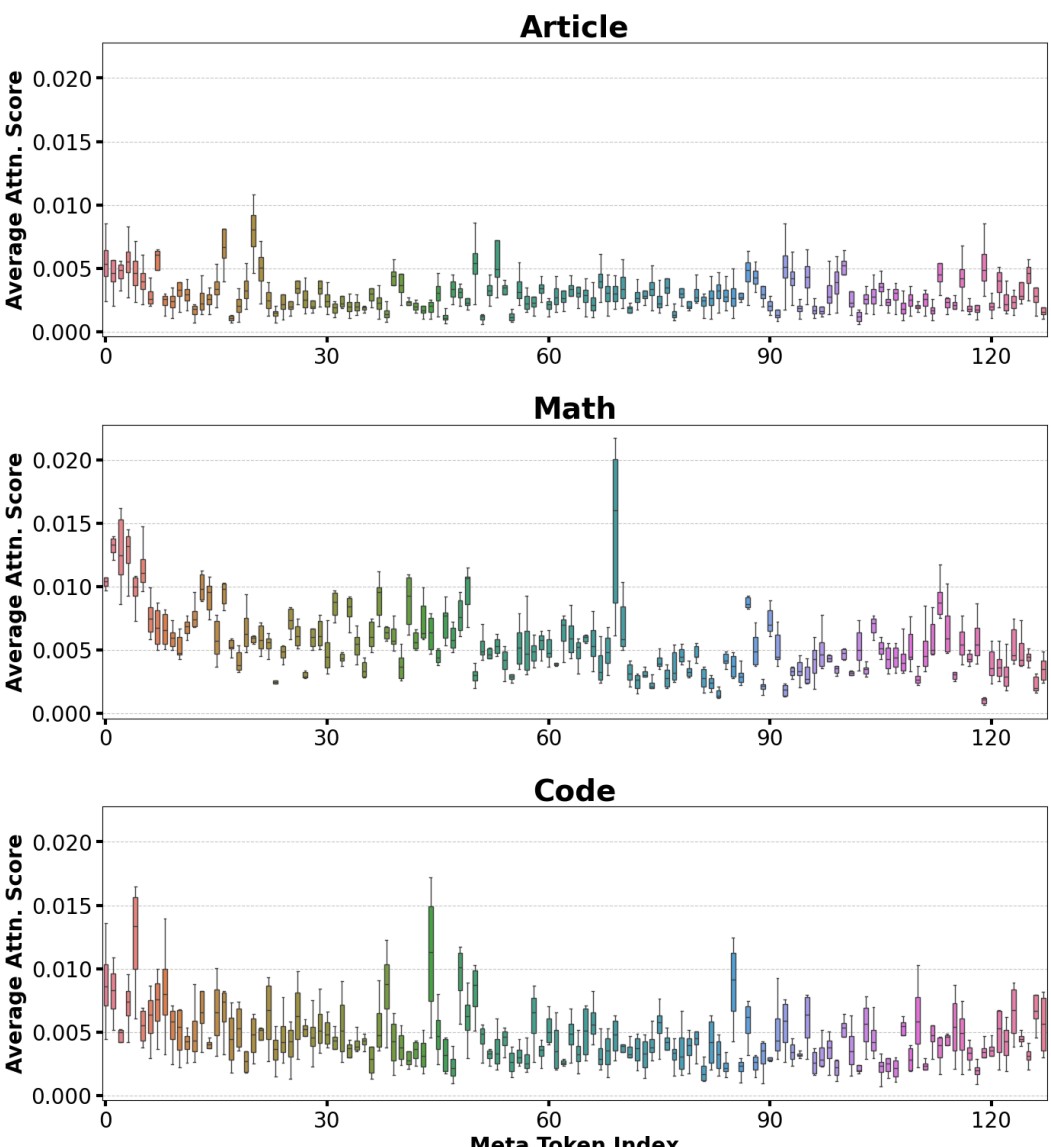

Figure 12: Averaged attention scores received by the meta tokens in the last layer of Hymba-1.5B model. Prompts of 'Article', 'Math' and 'Code' are from SQuAD (Rajpurkar et al., 2016), GSM8K (Cobbe et al., 2021), and GitHub-Code (CodeParrot) datasets, respectively.

with the findings in (Xiao et al., 2023). In addition, vanilla Transformers also have a higher proportion of 'Self' attention scores. In Hymba, meta tokens, attention heads and SSM heads work complimentary to each other, leading to a more balanced distribution of attention scores across different types of tokens. Specifically, meta tokens offload the attention scores from 'BOS', allowing the model to focus more on the real tokens. SSM heads summarize the global context, which focus more on current tokens (i.e., 'Self' attention scores). Attention heads, on the other hand, pay less attention to 'Self' and 'BOS' tokens, and more attention to other tokens (i.e., 'Cross' attention scores). This suggests that the hybrid-head design of Hymba can effectively balance the attention distribution across different types of tokens, potentially leading to better performance.

## G    META TOKENS: MORE ANALYSIS AND VISUALIZATION

**Interpretation from the memory aspect.** Similar to the analogy in Sec. 2.1, the meta tokens participate in the attention and SSM calculations of all subsequent tokens, analogous to metamemory

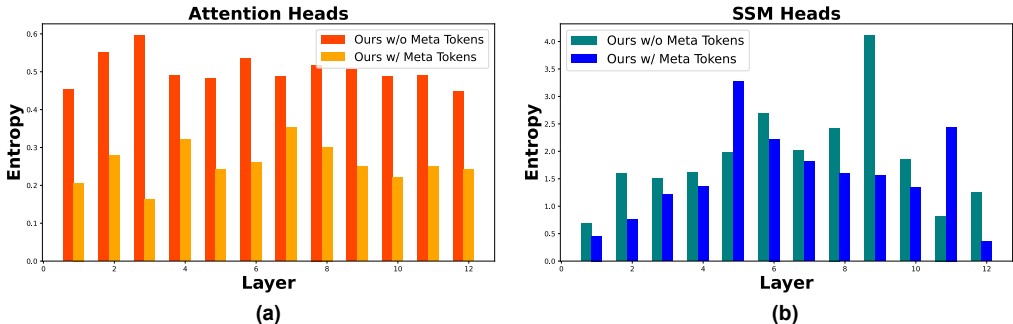

Figure 13: Visualize the layer-wise attention map entropy of (a) attention heads, and (b) SSM heads with and without meta tokens.

in the human brain, which helps recognize where to locate needed information in other memories. To see this, we visualize the averaged attention scores received by the meta tokens in Fig. 12 for a Hymba-1.5B model. We observe that when the prompts are from different domains (e.g., article, math, and codes), different meta tokens are activated. This suggests that different meta tokens encapsulate different world knowledge, which can be leveraged to guide the attention mechanism to focus on relevant information.

**Relationship with prior works.** Learnable tokens have also been leveraged in previous transformer-based models. Previous prompt tuning works (Lester et al., 2021; Gu et al., 2021c) prepend learnable prompts while keeping the model weights frozen during the task-specific tuning stage, aiming to adapt a pretrained LM to downstream tasks in a parameter-efficient manner. (Burtsev et al., 2020) introduces both learnable tokens and corresponding memory update modules to augment the memory mechanism in transformers. (Darcet et al., 2023) appends a set of learnable tokens called registers to the image patches of vision transformers (Dosovitskiy, 2020) to store global information and improve visual recognition. Our method integrates ideas from all these works in a more flexible manner. It jointly optimizes meta tokens with model weights during pretraining, remains compatible with SWA heads and other attention types or SSMs, and converts meta tokens into KV-cache initialization during inference without modifying the architecture.

**Meta tokens reduce attention map entropy.** We visualize the entropy of the attention map for both the attention and SSM heads (Ali et al., 2024; Ben-Kish et al., 2024) before and after introducing meta tokens. As introduced in Sec. 2.3 of our main paper, the attention map entropy reflects the distribution of attention scores across tokens, where lower entropy indicates stronger retrieval effects (Ren et al., 2024), as the attention scores are concentrated around a smaller subset of tokens.

As shown in Fig. 13, we observe that after introducing meta tokens, both the attention and SSM heads exhibit an overall reduction in entropy. Specifically, entropy is significantly reduced in all attention heads and in 10 out of 12 layers of the SSM heads. This suggests that meta tokens can reduce attention map entropy, potentially helping both the attention and SSM heads focus more on a subset of important tokens that contribute most to task performance, as indicated by the boosted performance in Tab. 11.

Table 12: Ablation study on the number of meta tokens.

| Model (300M) | Wiki. ppl. | LMB. ppl. | Lambda | PIQA | ARC-C | ARC-E | Hella. | Wino. | TruthfulQA | SIQA | Avg. |
|---|---|---|---|---|---|---|---|---|---|---|---|
| Hymba w/o meta tokens | 28.99 | 18.68 | 41.26 | 71.55 | 24.66 | 51.43 | 47.48 | 55.17 | 29.21 | 40.53 | 45.16 |
| Hymba w/ 128 meta tokens | 28.53 | 15.45 | 44.67 | 70.73 | 26.28 | 53.20 | 48.24 | 53.35 | 27.88 | 39.92 | 45.53 |
| Hymba w/ 256 meta tokens | 28.85 | 16.20 | 43.43 | 72.47 | 26.37 | 51.68 | 48.33 | 53.75 | 28.42 | 40.07 | 45.57 |

**Number of meta tokens.** To better understand the relationship between the number of meta tokens and the performance of the model, we further compare the performance of Hymba-300M with 0, 128, and 256 meta tokens, trained on 100B tokens from Fineweb (Penedo et al., 2024), following the apple-to-apple comparison in Tab. 8. As shown in the Tab. 12, we observe that (1) compared to Hymba without meta tokens, adding meta tokens consistently boosts the average accuracy and reduces the language model perplexity; (2) increasing the number of meta tokens from 128 to 256 does not result in a notable boost in average accuracy. As such, we adopt 128 meta tokens in Hymba.

Table 13: Architecture details of Hymba models of different size.

| Attribute | 125M | 350M | 1.5B |
|---|---|---|---|
| Blocks | 24 | 32 | 32 |
| Hidden Size | 512 | 768 | 1600 |
| SSM State | 16 | 16 | 16 |
| Attn. Heads | 8 | 12 | 25 |
| Query Groups | 4 | 4 | 5 |
| Num. Full Attn | 3 | 3 | 3 |
| Window Size | 1024 | 1024 | 1024 |
| MLP Hidden | 1664 | 2432 | 5504 |
| Tie Embedding | True | True | True |
| Parameters | 125M | 350M | 1.52B |

## H  PRETRAINING AND POST-TRAINING IMPLEMENTATION DETAILS

**Pretraining settings.** We train Hymba-125M/350M/1.5B models on 1.5T tokens, using a mix of DCLM-Baseline-1.0 (Li et al., 2024), SmolLM-Corpus (Ben Allal et al., 2024), and an internal high-quality dataset for 1T, 250B, and 50B tokens, respectively. We adopt the WSD learning rate scheduler (Hu et al., 2024) with three phases: (1) warmup steps set to 1% of the total steps, (2) a stable phase maintaining the peak learning rate of 3e-3, and (3) a decay phase reducing the learning rate to 1e-5 over 20% of the total steps, while gradually annealing to smaller, higher-quality datasets like SmolLM-Corpus and the internal dataset. We use a sequence length of 2K and a batch size of 2M tokens throughout the training process, which is conducted on 128 NVIDIA A100 GPUs. Details of Hymba-125M/350M/1.5B models are shown in Tab. 13. We also show the training curves of Hymba-1.5B in Fig. 14.

**Implementation details of post-training.** We post-train our 1.5B base model with a two-stage strategy: the first full-finetuning (FFT) stage and another DPO (Rafailov et al., 2024) training. The learning rates are 5e-5, and 3e-6 for FFT and DPO, respectively. Both FFT and DPO training are carried out for one epoch with a cosine scheduler. The global batch size is set to 1024. To accelerate training, we follow the training recipe (Tunstall et al., 2023; Diao et al., 2024; Dong et al., 2024) to pack the samples and use a block size of 2048. We implement the finetuning and DPO training with the LMFlow toolkit (Diao et al., 2024). In addition to full-finetuning, we also leverage Dora (Liu et al., 2024d) to do parameter-efficient finetuning.

**Baselines and downstream tasks.** We compare Hymba-1.5B-Instruct with competitive lightweight instruction-tuned models, including Llama-3.2-1B-Instruct (AI, 2024c), OpenELM-1-1B-Instruct (Mehta et al., 2024), Qwen2.5-1.5B-Instruct (Team, 2024), and SmolLM-1.7B-Instruct (Allal et al., 2024b). We evaluate these instruction-tuned models on MMLU (5-shot), IFEval, GSM8K (5-shot), GPQA (0-shot), and the Berkeley Function-Calling Leaderboard v2 (BF-CLv2) (Yan et al., 2024). For BFCLv2, we use the official code from the Gorilla project (Yan et al., 2024) and assess the BFCLv2-live category, which includes *live_simple*, *live_multiple*, *live_parallel*, *live_parallel_multiple*, and *live_relevance*. We exclude *live_irrelevance* since we found that some

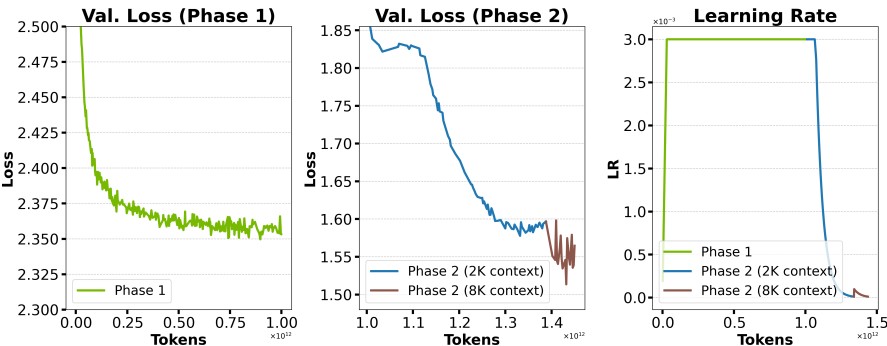

Figure 14: Training curves of Hymba-1.5B.

baseline models without function-calling capabilities achieved high scores in this category (where function calling is not required) but performed poorly on other tasks. As a result, these models attained high overall accuracy despite being ineffective for function calling. For the remaining tasks, we use the lm-evaluation-harness (Gao et al., 2024) for evaluation.

**Implementation of Hymba's forward process.** We provide illustrative pseudocode for Hymba's forward process in Alg. 1.

---

**Algorithm 1:** Forward Process of Hymba-1.5B

---

**Input**: $X = [x_1, \ldots, x_n]$, where $X \in \mathbb{R}^{(n,d)}$ are text input tokens.

**Model Configurations**:

- Number of blocks: 32

- Block indices with global attention: $[1, 16, 32]$ // Three global attention

- KV reusing groups: $[2, 3], [4, 5], [6, 7], [8, 9], [10, 11], [12, 13], [14, 15], [17, 18, 19],$ $[20, 21], [22, 23], [24, 25], [26, 27], [28, 29], [30, 31]$ // share KV per group

**Model Forward:**
$\tilde{X}^0 = [R, X] = [r_1, \ldots, r_m, x_1, \ldots, x_n]$ // Prepend $n$ meta tokens $R \in \mathbb{R}^{(m,d)}$
**for** *block-i* ***in*** $[1, \ldots, 32]$ **do**
   **if** *block-i* ***in*** $[1, 16, 32]$ **then**
      $\tilde{X}^i = \text{HYMBABLOCK-GA}(\tilde{X}^{i-1})$ // global attention
   **else**
      **if** *block-i* ***is the first block in its KV reusing group*** **then**
         $\tilde{X}^i, KV^i = \text{HYMBABLOCK-SWA}(\tilde{X}^{i-1})$ // sliding window attention
      **else**
         Retrieve KV cache from the previous layer: $KV^{i-1}$
         $\tilde{X}^i = \text{HYMBABLOCK-SWA}(\tilde{X}^{i-1}, KV^{i-1})$ // reuse KV
      **end**
   **end**
**end**

---

