# OpenReview forum: "Hymba: A Hybrid-head Architecture for Small Language Models"
_ICLR.cc/2025/Conference — ICLR 2025 Spotlight_

### Official Review · Reviewer_wExb · 2024-11-04

**Soundness:** 3
**Presentation:** 3
**Contribution:** 3
**Rating:** 8
**Confidence:** 5

**Summary:**

This paper introduces a new hybrid model named Hymba that integrates attention mechanisms with SSMs in a hybrid-head manner. The main difference between existing models like Samba is that Hymba enables hybrids to operate in parallel rather than sequentially.

The authors progressively propose several augmentations to the hybrid-head framework, including local/global attention, KV cache sharing, and meta tokens. They found that Hymba performs strongly against existing pure attention-based models and other hybrid models. By training the small-scale Hymba on trillions of tokens, Hymba performs well on common benchmarks and achieves near-perfect results on NIAH tests.

**Strengths:**

* This is a very solid work, with numerous experimental results and ablation studies verifying the effectiveness of Hymba, which is quite convincing.
* The results on NIAH tests are very impressive, especially given the model scales.

**Weaknesses:**

* I suggest the authors to add discussions with InfiniteTransformer, which fuses LA and attn in similar manners (Eq. 10)
* Why not the authors conduct experiments on Mamba2 rather than Mamba?
* If possible, I suggest the authors to add some discussions with more existing linear attention works like RetNet/GLA/HGRN2/YOCO

InfiniteTransformer: Efficient Infinite Context Transformers with Infini-attention

**Questions:**

* In Table 2, why are the ARC-C scores reported for 25 shots? I believe the common choice is zero shot.
* I am curious about how the throughputs in Table 2 are measured. Given that RWKV6 is reported to be much faster than others, this does not match my impressions. What is the input to the model? Can A100 GPUs with 80GB of memory handle an input size of 128 * 8K?

---

> ### Author Response · Authors · 2024-11-22
> **Response to Reviewer wExb**
>
> Thank you for your time and valuable feedback on our paper! We appreciate your recognition of “**the solidity of our work with numerous experimental results and ablation studies**”, “**impressive NIAH results given the model size**”, and “**performs strongly against existing pure attention-based models and other hybrid models**”.
>
>
> **Q1**: Discussions about InfiniteTransformer
> >  I suggest the authors to add discussions with InfiniteTransformer, which fuses LA and attn in similar manners (Eq. 10)
>
> Thank you for providing the reference. We humbly clarify that, although the Infinite Transformer (i.e., Infini-Attention) shares a similar concept of fusing different operators, methods and architectures of the two works are completely different. Infinite Transformer manually splits the input sequence into segments and performs segment-based memory updates, which is different from the end-to-end sequence processing in Hymba.
>
> More specifically, the Infinite Transformer splits the input sequence into several segments and processes each segment one by one using local quadratic attention and linear attention, referred to as compressive memory by the original paper, which is used to store information from past segments. This memory remains fixed while processing the current segment and is updated only after the segment is fully processed. This segment-based memory update process is distinct from our Hymba, which is an end-to-end model that updates the memory of its hybrid heads for each token rather than segment. This simplicity in design also makes Hymba easy for real-world deployment.
>
> Additionally, Hymba integrates other techniques for comprehensive optimization, such as meta tokens, global/local attention, and cross-layer KV sharing, making Hymba the strongest sub-2B model compared to small LM baselines. We will include these discussions in the final paper.
>
>
> **Q2**: Whether to adopt Mamba or Mamba2 in Hymba
> > Why not the authors conduct experiments on Mamba2 rather than Mamba?
>
> Hymba can work with either Mamba or Mamba2. We adopted Mamba based on our empirical results detailed below. Additionally, our finding is consistent with Jamba-1.5 [1]'s observation that, in Attention-Mamba hybrid models, Mamba often yields lower training loss and better performance compared to Mamba-2 (see Figure 1 in the Jamba-1.5 report).
>
> We further trained a 1B Hymba model with Mamba2 as its SSM heads on the SmolLM corpus using 100B data points and benchmarked it against other 1B models, following the apple-to-apple comparison setting in Table 3 of our submitted manuscript.
>
> As shown in the table below, we observed that (1) Hymba with Mamba heads performs better than Hymba with Mamba2 heads in terms of both language modeling and average commonsense reasoning accuracy; (2) Hymba with Mamba2 heads still outperforms other baseline 1B model architectures, demonstrating the general effectiveness of hybrid-head structures. This also suggests the potential for further performance enhancement with the advent of future advanced SSM operations, which can be integrated into Hymba.
>
> |  	| Language Modeling 	|  	|  	| Commonsense Reasoning 	|  	|  	|  	|  	|  	|  	|  	|  	|
> |---|---|---|---|---|---|---|---|---|---|---|---|---|
> | Model (1B) 	| WikiText (ppl.) 	| Lambda (ppl.) 	|  	| Avg. 	| Lambda 	| PIQA 	| ARC-C 	| ARC-E 	| Hellaswag 	| Winogrande 	| TruthfulQA 	| SIQA 	|
> | Mamba2 	| 19.17 	| 12.59 	|  	| 52.52 	| 47.51 	| 73.94 	| 38.91 	| 70.96 	| 57.73 	| 58.48 	| 30.75 	| 41.86 	|
> | LLaMA 	| 19.28 	| 13.09 	|  	| 52.82 	| 47.95 	| 73.45 	| 39.68 	| 73.74 	| 57.64 	| 56.20 	| 31.64 	| 42.22 	|
> | Samba 	| 19.91 	| 12.65 	|  	| 52.83 	| 49.08 	| 73.23 	| 39.59 	| 73.36 	| 58.49 	| 57.54 	| 28.84 	| 42.48 	|
> | Hymba (SSM=Mamba2) 	| 18.74 	| 11.58 	|  	| 53.31 	| 50.09 	| 74.27 	| 39.68 	| 72.18 	| 59.07 	| 57.62 	| 31.61 	| 41.97 	|
> | Hymba (SSM=Mamba) 	| 18.62 	| 10.38 	|  	| 54.57 	| 52.84 	| 74.97 	| 41.72 	| 74.12 	| 60.05 	| 57.85 	| 31.76 	| 43.24 	|
>
> [1] “Jamba-1.5: Hybrid Transformer-Mamba Models at Scale”, Jamba Team, arXiv’24.
>
>
> **Q3**: Discussions with more existing linear attention works like RetNet/GLA/HGRN2/YOCO
> > If possible, I suggest the authors to add some discussions with more existing linear attention works like RetNet/GLA/HGRN2/YOCO
>
> Thank you for providing the reference! In this work, we primarily focus on advancing the accuracy-efficiency frontier of small LMs through a hybrid-head structure, meta tokens, and cache optimization. The mentioned works represent more recent and advanced developments in linear attention, which can serve as plug-in linear attention (SSMs) within our hybrid-head structure, fusing with standard attention to further enhance achievable performance. As such, we believe these works and Hymba can mutually benefit from each other. We have added a brief discussion in Section 2.2 of our revised paper and will expand on it in the final paper.

---

> ### Author Response · Authors · 2024-11-22
> **Response to Reviewer wExb (Part 2)**
>
> **Q4**: Zero-shot ARC-C score
> >  In Table 2, why are the ARC-C scores reported for 25 shots? I believe the common choice is zero shot.
>
> We used 25-shot ARC-C in our submission to align with meta-llama/Llama-3.2-1B's evaluation setting on Huggingface [2], where they use 25-shot ARC-C.
>
> Following the reviewer's suggestion, we evaluated the 0-shot ARC-C and report the results below. Consistent with the 25-shot ARC-C results, our Hymba performs the best on 0-shot ARC-C compared to all sub-2B LMs.
>
> | Model           	| OpenELM-1 	| Llama-3.2-1B 	| Rene-v0.1 	| Phi-1.5 	| SmolLM 	| Cosmo 	| h2o-danube2 	| Hymba 	|
> |-----------------|-----------|--------------|-----------|---------|--------|-------|-------------|-------|
> | Size            	| 1.1B      	| 1.2B         	| 1.3B      	| 1.3B    	| 1.7B   	| 1.8B  	| 1.8B        	| 1.5B  	|
> | ARC-C (25-shot) 	| 33.87     	| 32.80        	| 36.95     	| 49.40   	| 46.67  	| 34.81 	| 40.61       	| 52.05 	|
> | ARC-C (0-shot)  	| 19.54     	| 31.39        	| 31.06     	| 44.71   	| 43.43  	| 32.94 	| 33.19       	| 45.90 	|
>
> [2] https://huggingface.co/meta-llama/Llama-3.2-1B#base-pretrained-models
>
>
> **Q5**: Throughputs measurement
> > I am curious about how the throughputs in Table 2 are measured.
>
> For throughput measurement, we use a batch size of 128 and a sequence length of 8k to evaluate batch generation efficiency. For models that encounter an Out-of-Memory (OOM) error, we halve the batch size until the OOM issue is resolved. This provides a measurement of the maximally achievable throughput without OOM, which is useful for efficient batch generation with memory constraints. Thank you for pointing this out and we have detailed this information to the revised manuscript.
>
> To further address your question, we have also provided throughput measurements with a batch size of 32 and a sequence length of 8k for all models. The results are shown in the table below, where the cache size is calculated based on an 8k sequence length, assuming an FP16 format, and the average accuracy is computed as the mean over the seven tasks reported in Table 2 of our submitted manuscript.
>
> We can observe that our Hymba-1.5B still achieves the best average accuracy among all models, along with better cache efficiency and throughput compared to pure transformer or other hybrid models. For example, compared to the strongest baseline, Llama-3.2-3B, trained with 9 trillion tokens, our Hymba-1.5B, trained with 1.5 trillion tokens, achieves a 1.26% improvement in average accuracy, 11.62x cache efficiency, and 2.21x throughput when measured with the small batch size of 32.
>
> In addition, regarding your question about RWKV6's throughput results, this is due to its use of a linear attention formulation, which is computationally and memory-efficient compared to quadratic attention, particularly with the adopted 8k sequence length. In comparison, our Hymba achieves significantly higher average accuracy (+11.97%) than RWKV6, thanks to the proposed hybrid model design.
>
>
> |  | #Params. | Model Type| Cache Size (MB) | Throughput (tok/sec) | Average Acc (%) |
> |---|:---:|:---:|:---:|:---:|:---:|
> | Rene-v0.1 | 1.3B | Hybrid | 113 | 800 | 51.68 |
> | RWKV6 | 1.6B | Linear Attention| 6 | 927 | 47.32 |
> | Phi-1.5 | 1.3B | Transformer| 1573 | 241 | 53.65 |
> | SmolLM | 1.7B | Transformer| 1573 | 238 | 52.78 |
> | Cosmo | 1.8B | Transformer| 1573 | 244 | 45.59 |
> | h2o-danube2 | 1.8B | Transformer| 492 | 259 | 53.95 |
> | Llama-3.2 | 3.0B | Transformer| 918 | 191 | 58.11 |
> | Hymba | 1.5B | Hybrid | 79 | 423 | 59.37 |

---

> > ### Comment · Reviewer_wExb · 2024-11-23
> >
> > Thank you for your hard work on this manuscript.
> > I believe it greatly benefits from the extended discussions and experimental results.
> > Overall, this is a very solid work, so I have increased my score to 8.

---

> > > ### Author Response · Authors · 2024-11-23
> > > **Response to Reviewer wExb**
> > >
> > > Thank you for recognizing the solidity of our work! Following your suggestions, we will include the discussions about Infinite Transformer and other linear attention works in our final version and add the above ablation studies to the appendix.

---

### Official Review · Reviewer_nHjx · 2024-11-04

**Soundness:** 3
**Presentation:** 2
**Contribution:** 3
**Rating:** 6
**Confidence:** 3

**Summary:**

This paper introduces Hymba, a hybrid architecture that combines Transformer and SSM within a single layer. The authors also propose several additional techniques to enhance further efficiency and performance, such as combining global and local attention, cross-layer KV cache sharing, and introducing meta-tokens, which act as a learned prefix. Through extensive evaluation, the authors show that Hymba performs best among small LMs while being significantly more efficient.

**Strengths:**

1. The paper introduces several useful designs that could be empirically used for training hybrid Transformer + SSM models.
2. The proposed method shows high performance, outperforming most small LMs while achieving better computation efficiency.
3. The authors perform extensive evaluations across diverse tasks and setups.a

**Weaknesses:**

1. Limited novelty. The paper seems to suggest a combination of implementation tricks rather than proposing a significant idea.
2. The hybrid head design seems to be the most significant component of the proposed method, but the evaluation justifying its efficacy is confusing. In Figure 3, the authors compare their method with Samba and claim they achieved a larger ERF. However, it is unclear if the gain comes from the parallel design or the introduction of global attention heads (not present in Samba).

**Questions:**

1. Does the Samba baseline in Figure 3 also use the same number of global attention layers? If not, how can we tell if the performance gain comes from the parallel design or the introduction of global attention layers?
2. How does the parallel design impact the throughput? Empirically, are the SSM heads and Attention heads computed in parallel or sequentially? (e.g., if you forward the input through the SSM heads, then forward the input through the attn heads, and then aggregate them, then the implementation is done sequentially, even if the design is conceptually ‘parallel’) Would ‘true’ parallel computation require a specialized GPU kernel?
3. Is the concept of meta-tokens useful for SSMs only, or would general Transformer models also benefit from the technique?

---

> ### Author Response · Authors · 2024-11-22
> **Response to Reviewer nHjx**
>
> We sincerely thank the reviewer for the recognition of strengths including "**several useful designs**", "**high performance**", "**outperforming most small LMs while achieving better computation efficiency**", and "**extensive evaluations across diverse tasks and setups**". We try to address your constructive comments in the following.
>
>
> **Q1**: Limited novelty
> > Suggest a combination of implementation tricks rather than proposing a significant idea
>
>
> As pointed out by other reviewers as well, e.g., “an innovative approach” by Reviewer HFsT and “very solid work” by Reviewer wExb, we humbly clarify that our work delivers rich new contributions through its key features: (1) To our knowledge, we are the first to propose and thoroughfully explore the hybrid-head structure for LMs, demonstrating the remarkable effectiveness of parallelly processing the same input through hybrid operators; (2) The proposed learnable meta tokens effectively alleviate the “force-to-attend” issue and reduce high attention scores on semantically unimportant tokens, as shown in the attention map visualization in Figure 8 in Appendix C of the revised manuscript; (3) The hybrid building blocks integrated in Hymba enable extensive use of local attention mechanisms while maintaining high recall accuracy, as evidenced by the ablation study in Table 9 in the appendix.
>
> Furthermore, in addition to the aforementioned new techniques/insights, the strong performance achieved by Hymba further validates and justifies our contributions. For example, notably, Hymba-1.5B, trained on only 1.5T tokens, stands out as the strongest sub-2B model among existing sub-2B LMs, demonstrating the efficacy and significance of the above architectural innovations. As such, we can expect that the Hymba architecture and its pre-trained models, which will be open-sourced, can significantly advance the frontier of edge LMs and inspire further innovations in LMs.
>
> Finally, beyond the novel modules, insights, and strong performance, we would like to emphasize that comprehensive evaluation, ablation studies, and analysis are crucial for community development especially when it comes to LMs. We have provided a detailed design roadmap and analysis in Table 1 (along with additional ablation studies in Table 9) of our submitted manuscript to offer design insights and implementation guidelines, thereby facilitating fair benchmarks and inspiring future small LMs.

---

> ### Author Response · Authors · 2024-11-22
> **Response to Reviewer nHjx (Part 2)**
>
> **Q2**: More comparison to Samba
> > Does the Samba baseline in Figure 3 also use the same number of global attention layers?
> > how can we tell if the performance gain comes from the parallel design or the introduction of global attention layers?
>
>
> We humbly clarify that Samba's original architecture does not include global attention layers according to their paper and codes. As such, in our apple-to-apples comparison in Tables 3 and 8, we followed the original Samba design to avoid confusion.
>
> To address the reviewer’s question, we further built a variant of Samba where we replaced its first, last, and middle local attention layers with global attention layers to ensure this variant also has 3 global attention layers, which is the same strategy as our Hymba model. We call this variant Sequential-Mix-Attention (SMA), which is used to study the relative contributions of the parallel design and mixed global/local attention.
>
>
> We conducted apple-to-apples comparisons among Hymba, Samba, and the new SMA and reported their performance in the following.
>
> | Task        	| Samba-300M 	| SMA-300M 	| Hymba-300M 	|
> |-------------------|----------------------------|--------------------|----------------------------|
> | Wiki. ppl.  	| 31.41      	| 29.75    	| 28.53      	|
> | LMB. ppl.   	| 19.75      	| 20.85    	| 15.45      	|
> | SQuAD-C     	| 39.88      	| 44.44    	| 45.24      	|
> | SWDE        	| 22.14      	| 55.48    	| 58.33      	|
> | Avg.        	| 31.01      	| 49.96    	| 51.79      	|
> | Lambda      	| 40.59      	| 40.40    	| 44.67      	|
> | PIQA        	| 69.86      	| 69.80    	| 70.73      	|
> | ARC-C       	| 25.76      	| 25.94    	| 26.28      	|
> | ARC-E       	| 49.79      	| 49.62    	| 53.20      	|
> | Hella.      	| 46.45      	| 46.42    	| 48.32      	|
> | Wino.       	| 52.49      	| 52.72    	| 53.35      	|
> | TruthfulQA  	| 27.27      	| 26.47    	| 27.87      	|
> | SIQA        	| 39.92      	| 41.25    	| 39.92      	|
> | Avg.        	| 44.02      	| 44.08    	| 45.53      	|
>
> These results reinforce our contributions in the paper:
> 1. The parallel design (Hymba) outperforms its sequential counterpart (SMA) in benchmarks, including perplexity, recall-intensive tasks, and commonsense-reasoning & QA.
>
>     This aligns with the results in Table 1 of the submitted manuscript, where we also compared hybridizing global attention and SSM in a sequential way (i.e., "A. + Attention heads (sequential)") with hybridizing global attention and SSM in a parallel way (i.e., "B. + Multi-head structure (parallel)") and found that the latter performs considerably better.
>
> 2. Although SMA does not show superior performance over Samba on commonsense-reasoning & QA tasks, it outperforms Samba in recall-intensive tasks (SQuAD-C and SWDE). This reflects one of our contributions regarding mixing local and global attention, which we discussed in Section 2.3.
>
> In summary, the parallel design in our Hymba contributes to improved accuracy in general commonsense reasoning tasks, while the mixed global/local attention ensures high recall accuracy. We will include this discussion in our final version.
>
>
> **Q3**: The achievable efficiency of the parallel design
> > How does the parallel design impact the throughput?
> > Would ‘true’ parallel computation require a specialized GPU kernel?
>
> Thank you for the good question!
> You are correct that true parallel computation requires a specialized GPU kernel, which can further improve the achievable throughput performance than our reported ones. In our current implementation, the SSM heads and attention heads are computed sequentially because we are using HuggingFace/Transformers’ available modules to implement the model for ease of use and compatibility with other frameworks. As such, the improved throughput over transformer-based models reported in Table 2 of our submitted manuscript stems from optimized cache efficiency and reduced computation, rather than parallel execution. Hence, we can expect even stronger throughput performance  for Hymba when true parallel computation is adopted.
>
> Hymba’s parallel design improves accuracy over sequential designs like Samba (as addressed in Q2), while also having the potential of achieving higher efficiency under true parallel execution. As you correctly point out, the latter requires a specialized GPU kernel. We are actively working on this kernel, aiming to unleash the full potential of Hymba’s parallel design. For example, two CUDA streams could be used to execute SSM heads and attention heads in parallel, and the CUDA graph could be further optimized by integrating with deployment frameworks like vLLM. We will release it to the community once it is fully finished.

---

> > ### Comment · Reviewer_EGZg · 2024-11-26
> > **new baseline**
> >
> > thanks for adding this baseline, and the results are very solid. I have raised my score to 8 reflecting this.

---

> ### Author Response · Authors · 2024-11-22
> **Response to Reviewer nHjx (Part 3)**
>
> **Q4**: Effectiveness of Meta tokens for Transformers
> > Would general Transformer models also benefit from the technique?
>
> Thanks for the insightful question! Yes, we find that general transformer models also benefit from learnable meta tokens, in addition to their effectiveness on our hybrid model and pure Mamba, as provided in Table 9 of our submitted manuscript.
>
> To demonstrate this, we prepend 128 meta tokens to Llama3-1B and train the model from scratch on 100B data from the SmolLM corpus, following the apple-to-apples settings in Table 3 of our submitted manuscript.
>
> As shown in the table below, introducing meta tokens to Llama3-1B maintains comparable commonsense reasoning accuracy while notably boosting recall accuracy by 7.15%. This aligns with our analysis regarding the roles of meta tokens and the visualization of the attention map with meta tokens in Figure 8 in Appendix C: meta tokens guide the attention mechanism of subsequent tokens to focus on more important content and reduce the attention score on semantically unimportant tokens like the bos token.
>
> |  | Commonsense Reasoning Acc (%) | Recall Acc (%) |
> |---|:---:|:---:|
> | Llama3 | 52.82 | 47.33 |
> | Llama3 + Meta tokens | 52.68 | 54.48 |
>
> \* Commonsense reasoning accuracy is averaged over eight tasks, and recall accuracy is averaged over two tasks, following the settings in Table 3 of our submitted manuscript.

---

> > ### Comment · Reviewer_EGZg · 2024-11-26
> >
> > does this mean that meta-tokens helped more with the recall?

---

> > > ### Author Response · Authors · 2024-11-27
> > > **Further Response to Reviewer EGZg**
> > >
> > > Thank you for recognizing the solid results of our work! Following your suggestion, we will include the long-context results and additional architecture ablation in our final version.
> > >
> > > Regarding your question about whether meta-tokens help more with recall, we observe that (1) for models with Mamba blocks (Hymba/Mamba), meta-tokens can improve both commonsense reasoning accuracy and recall accuracy, as shown in Table 1 Row-E for Hymba and Table 10 Row-12 for Mamba; (2) for transformer models, meta-tokens primarily enhance recall accuracy by guiding the attention mechanism to focus more on semantically important tokens, as demonstrated by the visualization of the attention map with meta-tokens in Figure 8 in Appendix C.

---

> ### Comment · Reviewer_nHjx · 2024-11-26
>
> Thank you for your efforts. I appreciate the clarification on the Samba baselines and the additional experiments on SMA, as well as the Meta token evaluation on Transformers and highlighting the significance of this work. I am convinced that Hymba will be a valuable contribution to the community, and therefore, I am raising my score.

---

> ### Author Response · Authors · 2024-11-27
> **Further response to Reviewer nHjx**
>
> Thank you for your insightful suggestions regarding the SMA baseline and the parallel execution of our Hymba architecture! We will include the ablation study of the SMA design in the final version and actively explore further speedups for Hymba with fused parallel kernels.

---

### Official Review · Reviewer_EGZg · 2024-11-04

**Soundness:** 3
**Presentation:** 3
**Contribution:** 3
**Rating:** 8
**Confidence:** 3

**Summary:**

This paper introduces Hymba,  a small language model that combines attention mechanisms with SSMs in a hybrid-head architecture. Authors did the following,

1. A hybrid-head architecture that processes inputs through parallel attention and SSM heads in each layer, leveraging attention's high-resolution recall and SSM's efficient context summarization

2. Learnable meta tokens prepended to input sequences that act as learned cache initialization to modulate token processing

3. Optimization techniques including local/global attention combination and cross-layer KV sharing to improve efficiency

The authors validate their approach through extensive experiments showing Hymba1.5B achieves comparable performance to larger models while being 3x faster and using 15x less cache yielding memory gain.

**Strengths:**

1. Empirical study is comprehensive and clear, particularly the ablation studies;

2. Consistent gains for models with different sizes under 1.5B

3. The paper is easy to follow

**Weaknesses:**

In general I think this is a strong paper. I have the following comments and questions.

- Some implementation details can be added
1. I am a bit lost while reading the cache optimization, and meta-token, maybe worth more explanations or pseudocode?

- How many meta tokens are needed, and how they are related to the performance in downstream tasks?

- The ratio between SSM and Attention is not clear. And I understand that recent papers demonstrate that it is important to integrate attention for linear RNN models, but attention layer still added overhead, though coupled with all those techniques. A fair comparison could be a pure attention model with all methods proposed in this paper, comparing their efficiency gain and performance curve.

**Questions:**

1. The interplay between SSM and Attention, see weakness

2. Consider a more fair comparison for efficiency and performance gain? see weakness

3. How does the model perform for long context tasks as this is the where the gain of Hymba goes significant?

---

> ### Author Response · Authors · 2024-11-22
> **Response to Reviewer EGZg**
>
> Thank you for your time and comments. We greatly appreciate your recognition of our paper's highlights, including the "**comprehensive and clear ablation studies**", "**consistent gains**", and your description of it as an "**easy-to-follow and strong paper**". We address your comments and questions below.
>
>
> **Q1**: Some implementation details
> > the cache optimization, and meta-token, maybe worth more explanations or pseudocode?
>
> Sure, we are happy to elaborate further on the implementation details. Following your suggestion, we have added pseudocode for Hymba’s forward process in Appendix F of our revised manuscript. Additionally, we have clarified the details of cache optimization and meta tokens below and will include these clarifications in our final version.
>
> **Cache Optimization:** We use global attention in only three layers (the first, last, and middle) and employ sliding window attention (a.k.a. local attention) in all remaining layers. Furthermore, we group every two consecutive sliding window attention layers into a KV cache-sharing group. Only the first layer in each group computes and stores KV cache for tokens, while the second layer retrieves the stored KV cache and uses them to compute attention.
>
> **Meta Tokens:** After the embedding layer, the size of the text input tokens is \((n, d)\), where \(n\) is the sequence length and \(d\) is the model dimension. The size of the meta tokens is \((m, d)\), where \(m\) is the number of meta tokens. Meta tokens are prepended to the text input tokens, resulting in a \((m+n, d)\) matrix, which is fed to the model and learned jointly with the model weights during training.
>
> Additionally, we modify the attention mask to an “A-shape” pattern to ensure that sliding window attention can always attend to meta tokens, as illustrated in Figure 10 of the revised manuscript.
>
> We will provide pseudocode in the updated manuscript and release our implemenation codes and models.
>
>
>
> **Q2**: Ratio of SSM and Attention
> >  The ratio between SSM and Attention is not clear.
>
> Thank you for the constructive feedback. As shown in Table 9 of the Appendix in our submitted manuscript, we studied the relationships among three factors: the ratio between Attention and Mamba, model performance (i.e., general and recall-intensive tasks), and efficiency (i.e., throughput and cache). These factors, along with other design elements, are interrelated, and we provided detailed ablation studies under various settings.
>
> Generally, we observed that model performance improves as the ratio of attention parameters increases, although this improvement gradually saturates. The resulting architecture we adopt has approximately a 1:5 Attention-to-Mamba parameter ratio, which achieves a good balance between performance and efficiency.

---

> ### Author Response · Authors · 2024-11-23
> **Response to Reviewer EGZg (Part 2)**
>
> **Q3**: The number of meta tokens
> > How many meta tokens are needed, and how they are related to the performance in downstream tasks
>
> In our submission, we add 128 meta tokens to Hymba. This is because we are using FlexAttention to support the attention mask (see Figure 10 of our revised manuscript) during training, and FlexAttention prefers block sizes that are multiples of 128 for the attention mask.
>
> To better understand the relationship between the number of meta tokens and model performance, we further compare the performance of Hymba-300M with 0, 128, and 256 meta tokens, trained on Fineweb 100B, following the apple-to-apple comparison in Table 8 of our submitted manuscript.
>
> As shown in the table below, we observe that (1) compared to Hymba without meta tokens, adding meta tokens consistently boosts the average accuracy and reduces the language model PPL (-3.23/-2.48 on Lambda for 128/256 meta tokens, respectively); (2) increasing the number of meta tokens from 128 to 256 does not result in a notable boost in average accuracy. As such, we adopt 128 meta tokens in our Hymba design for simplicity.
>
> Additionally, an intriguing future work is to interleave normal input tokens and meta tokens, which allows meta tokens to summarize previous input tokens and further scale up. We will share our results with the community once they are ready.
>
>
> |  | Language Modeling PPL |  | Task Acc (%) |  |  |  |  |  |  |  |  |
> |---|:---:|:---:|:---:|:---:|:---:|:---:|:---:|:---:|:---:|:---:|:---:|
> | Model (300M) | WikiText | Lambda | Avg. | Lambda | PIQA | ARC-C | ARC-E | Hellaswag | Winogrande | TruthfulQA | SIQA |
> | Mamba | 30.78 | 19.95 | 42.98 | 38.95 | 69.64 | 24.91 | 50.67 | 44.95 | 51.70 | 23.86 | 39.20 |
> | Llama3 | 30.04 | 20.53 | 44.08 | 40.15 | 70.29 | 24.83 | 50.42 | 45.69 | 52.64 | 28.97 | 39.66 |
> | Hymba w/o meta tokens | 28.99 | 18.68 | 45.16 | 41.26 | 71.55 | 24.66 | 51.43 | 47.48 | 55.17 | 29.21 | 40.53 |
> | Hymba w/ 128 meta tokens | 28.53 | 15.45 | 45.53 | 44.67 | 70.73 | 26.28 | 53.20 | 48.24 | 53.35 | 27.88 | 39.92 |
> | Hymba w/ 256 meta tokens | 28.85 | 16.20 | 45.57 | 43.43 | 72.47 | 26.37 | 51.68 | 48.33 | 53.75 | 28.42 | 40.07 |
>
> We will respond to your remaining questions and comments soon.

---

> ### Author Response · Authors · 2024-11-24
> **Response to Reviewer EGZg (Part 3)**
>
> **Q4**: A fair comparison: pure attention model with all proposed cache optimization methods
> > Pure attention model with all KV cache optimization methods proposed in this paper
>
> Thank you for your suggestion! We have included the results of applying our KV optimization technique to pure transformers in Table 9 of our submitted manuscript.
>
> To make it clearer, we have also reorganized the results to provide a fair comparison among model architectures in the table below. Specifically, all models are 300M and trained on Fineweb 100B, following the settings in Table 9 of our submitted manuscript. “Llama3 + KV optim.” refers to the Llama3 model with our mixed global/local attention (a total of three global attention layers, the same as in our Hymba) and cross-layer KV sharing applied.
>
>
> | Model (300M) | WikiText PPL | Commonsense Reasoning Acc (%) | Recall Acc (%) | Cache (MB) | Throughput (tok/sec) |
> |:---:|:---:|:---:|:---:|:---:|:---:|
> | Mamba | 30.78 | 42.98 | 19.23 | **1.9** | **4720.8** |
> | Llama3 | 30.04 | 44.08 | 39.98 | 829.4 | 721.1 |
> | Llama3 + KV optim. | 31.50 | 43.61 | 28.18 | 56.6 | 3710.0 |
> | Hymba w/o meta tokens | 28.99 | 45.16 | 48.04 | 76.3 | 2756.5 |
> | Hymba w/ meta tokens | **28.53** | **45.53** | **51.79** | 76.9 | 2695.8 |
>
> **The commonsense reasoning accuracy is averaged over eight tasks, and the recall accuracy is averaged over two tasks. The throughput and cache size are measured using the settings in Table 9 of our submitted manuscript.*
>
>
> We observe the following:
>
> 1. After applying our KV cache optimization techniques to the pure transformer Llama3, the cache efficiency and throughput are indeed improved but at the cost of a +1.46 PPL increase, a 0.47% reduction in commonsense reasoning accuracy, and a significant 11.80% reduction in recall accuracy due to the lack of global context, compared to the vanilla Llama3.
>
>     In contrast, both Hymba models (with or without meta tokens) achieve >1.5% commonsense reasoning accuracy improvements and >19% recall accuracy improvements compared to this KV-optimized Llama3 model. As analyzed in Appendix B of our submitted manuscript, this is because the presence of SSM heads in our hybrid-head module effectively summarizes the global context, allowing us to more aggressively reduce the KV cache used to record the context. Conversely, aggressively reducing the KV cache for pure transformers may not be feasible.
>
> 2. Compared to the strongest baseline, Llama3, Hymba without meta tokens already achieves better language modeling (-1.05 PPL), better commonsense reasoning accuracy (+1.08%), and better recall accuracy (+8.06%), while achieving 3.82x throughput and 10.87x cache efficiency. With meta tokens, task performance is further improved while maintaining efficiency. This indicates that Hymba can more effectively achieve a better accuracy-efficiency trade-off compared to simply optimizing the KV cache of Llama3.

---

> ### Author Response · Authors · 2024-11-24
> **Response to Reviewer EGZg (Part 4)**
>
> **Q5**: Evaluation on long-context tasks
> > How does the model perform for long context tasks
>
> In Figure 5 and Section 3.3 of our submitted manuscript, we provided the Needle-in-a-Haystack (NIAH) evaluation across different model architectures under an apple-to-apple comparison setting, where we find that Hymba achieves better NIAH results compared to pure Transformer (Llama3) and Mamba. Additionally, we have considered recall-intensive tasks (SQuAD-Completion and/or SWDE) in Tables 2 and 3 of our submitted manuscript to demonstrate Hymba’s improved recall accuracy.
>
> To further address your question, we have also evaluated Hymba on more types of long-context tasks, including summarization and few-shot learning from LongBench [1]. Specifically, given the rebuttal time, we fine-tuned our Hymba-1.5B model on 8k context length using 50B data from the SmolLM corpus and benchmarked our model against the best-performing models in Table 2 of our submitted manuscript, i.e., h2o-danube2-1.8B (trained on 16k context length) and SmolLM-1.7B (trained on 2k context length). We evaluated all models on three English summarization tasks and four few-shot learning tasks from LongBench.
>
> As shown in the table below, Hymba performs the best across both types of tasks, even outperforming h2o-danube-1.8B, which has mcuh larger KV cache size and was trained on a longer context length. We also note that Hymba's long-context performance can be further improved by fine-tuning on longer sequences, which will be our focus in future release.
>
>
> |  | Summarization |  |  | Few Shot |  |  |  |
> |---|:---:|---|---|:---:|---|---|---|
> | Model | GovReport (Rouge-L) | MultiNews (Rouge-L) | QMSum (Rouge-L) | TriviaQA (F1) | SAMSum (Rouge-L) | TREC (Acc) | LSHT (Acc) |
> | SmolLM-1.7B | 4.77 | 12.79 | 8.55 | 1.97 | 3.23 | 1.00 | 0.00 |
> | h2o-danube-1.8B | 12.41 | 14.28 | 17.01 | 68.24 | 11.46 | 56.00 | 10.50 |
> | Hymba-1.5B | 13.95 | 19.24 | 17.29 | 76.82 | 35.21 | 56.22 | 11.00 |
>
> [1] Bai, Y., Lv, X., Zhang, J., Lyu, H., Tang, J., Huang, Z., Du, Z., Liu, X., Zeng, A., Hou, L. and Dong, Y., 2023. Longbench: A bilingual, multitask benchmark for long context understanding. arXiv preprint arXiv:2308.14508.

---

### Official Review · Reviewer_HFsT · 2024-11-07

**Soundness:** 4
**Presentation:** 4
**Contribution:** 3
**Rating:** 8
**Confidence:** 3

**Summary:**

The paper introduces **Hymba**, a new family of small language models designed with a hybrid-head architecture that merges attention mechanisms with state space models (SSMs) for improved memory functions. Hymba utilizes attention heads for precise recall of high-resolution information and SSM heads to summarize broader context efficiently, mirroring aspects of human memory. A significant innovation is the introduction of learnable meta tokens, which act as a dynamic cache initialization, enhancing focus on key information during inference.

The authors outline a systematic approach to developing Hymba, from creating fused hybrid modules to scaling model and data sizes. Experimental results show that Hymba sets new benchmarks for small language models, achieving an improved accuracy-efficiency balance. Notably, *Hymba-1.5B* matches the commonsense reasoning performance of larger models, such as *LLaMA 3.2 3B*, while operating more efficiently. The meta tokens also reduce attention map entropy, potentially aiding the model in identifying and focusing on salient tokens. Hymba’s design offers promising advances in both the performance and efficiency of compact language models.

**Strengths:**

1. this is a solid and well-written paper.
2. the hybrid-head design, which combines attention and state space models, is an innovative approach that provides Hymba with both fine-grained recall and efficient long-range summarization. The introduction of learnable meta tokens as a dynamic cache initialization mechanism is also novel, drawing a parallel to human metamemory functions.
3. the experiments are extensive and well-documented, including ablation studies that thoroughly evaluate the impact of each component, such as the hybrid heads and meta tokens. The benchmarks are comprehensive and competitive, providing a robust demonstration of Hymba's capabilities.

**Weaknesses:**

1. It would be even better if the effectiveness of the Hymba could be validated on image or speech modalities.

**Questions:**

1. Equation 1 does not mention a scaling factor. Is it included in the actual implementation?

---

> ### Author Response · Authors · 2024-11-21
> **Response to Reviewer HFsT**
>
> We sincerely thank the reviewer for their time and constructive comments! We appreciate the reviewer's recognition of our paper, including “**outlining a systematic approach to developing Hymba**”, “**the hybrid-head designs as an innovative approach**”, “**the significant innovation of introducing learnable meta tokens**”, “**extensive and well-documented experiments and ablation studies**”, resulting in “**a solid and well-written paper**”.
>
>
> **Q1**: Extend to other modality
> > It would be even better if the effectiveness of the Hymba could be validated on image or speech modalities.
>
>
> Thank you for the suggestion! Given the state-of-the-art performance achieved by Hymba among all sub-2B models, we believe that Hymba has high potential for different modalities, especially considering that Vision-Language Models (VLMs) and other multimodal foundation models are typically fine-tuned from pre-trained language models [1,2,3]. We are currently working on a Hymba-based VLM as future work of this submission and look forward to sharing our results with the community once they are available.
>
> [1] Liu, H., Li, C., Wu, Q. and Lee, Y.J., 2024. Visual instruction tuning. Advances in neural information processing systems, 36.
>
> [2] Awadalla, A., Gao, I., Gardner, J., Hessel, J., Hanafy, Y., Zhu, W., Marathe, K., Bitton, Y., Gadre, S., Sagawa, S. and Jitsev, J., 2023. Openflamingo: An open-source framework for training large autoregressive vision-language models. arXiv preprint arXiv:2308.01390.
>
> [3] Lin, J., Yin, H., Ping, W., Molchanov, P., Shoeybi, M. and Han, S., 2024. Vila: On pre-training for visual language models. In Proceedings of the IEEE/CVF Conference on Computer Vision and Pattern Recognition (pp. 26689-26699).
>
> **Q2**: Scaling factor in Equation 1 (Softmax Attention)
> > Equation 1 does not mention a scaling factor. Is it included in the actual implementation?
>
>
> Thank you for pointing this out. Yes, the scaling factor $\frac{1}{\sqrt{d}}$ is included in the actual implementation. We omit this for simplicity of illustration and have added a note in the revised paper to avoid confusion.

---

> > ### Comment · Reviewer_HFsT · 2024-11-24
> > **Official Comment by Reviewer HFsT**
> >
> > Thank you for your response. I have no further questions. I will keep my positive score.

---

> ### Author Response · Authors · 2024-11-24
> **Further response to Reviewer HFsT**
>
> Thank you for recognizing the innovation and extensive evaluation of our work! Following your suggestion, we will share our results on extending to new modalities with the community once they are ready.

---

### Author Response · Authors · 2024-11-24
**Thank All Reviewers and General Response**

We sincerely appreciate all reviewers for their recognition and constructive comments! We have addressed all the suggestions and concerns raised by the reviewers through additional experiments, clarifications, and extended content, with responses posted separately for each reviewer. Please let us know if you have further questions or suggestions and we would be happy to discuss them further.

Additionally, to help reviewers recall the details of our paper, we have summarized its content and listed the strengths highlighted by the reviewers below for your reference:

**Summary:**

In this submission, we present our innovations in Hymba, including Fused Hybrid Heads, Meta Tokens, and KV optimization via mixed local/global attention plus cross-layer KV cache sharing, supported by an extensive empirical design roadmap, ablation studies, and analyses to understand how and why each design works. Finally, we scale our findings to 1.5B parameters and deliver state-of-the-art LLM models in their scale category, achieving an improved balance between accuracy and efficiency. Additionally, we will provide a fully open version of Hymba to facilitate future innovations within the community.

**Strengths commented by the reviewers:**

*[Innovative Designs]*

Reviewer HFsT: The hybrid-head design is an innovative approach. The introduction of learnable meta tokens is also novel.

Reviewer nHjx: The paper introduces several useful designs that could be empirically used for training hybrid Transformer + SSM models.

*[Strong Performance]*

Reviewer HFsT: The experiments are extensive and well-documented.

Reviewer EGZg: Consistent gains for models with different sizes under 1.5B.

Reviewer nHjx: Outperforming most small LMs while achieving better computation efficiency.

Reviewer wExb: Results on NIAH tests are very impressive, especially given the model scales.

*[Solid Ablation Results]*

Reviewer EGZg: Empirical study is comprehensive and clear, particularly the ablation studies.

Reviewer wExb: This is a very solid work, with numerous experimental results and ablation studies verifying the effectiveness of Hymba, which is quite convincing.

*[Presentations]*

Reviewer HFsT: this is a solid and well-written paper.

Reviewer EGZg: The paper is easy to follow.

---

### Meta-Review · Area_Chair_jKAo · 2024-12-18

**Metareview:**

This paper proposes a new hybrid approach of Transformer LMs and state-space models.
To overcome the limitiation of Transformers is very important and practical, this paper designs an effective hybrid approach with extensive experimental evidence.
Because there exist multiple hybrid efforts of SSM and Attention methods, the fundamental novelty might be not strong. But, considering the difficulty of effective integration of two apporaches, reviewers and AC respect the contributions of the proposed method design.

All reviewers gave positive ratings and AC also agree with their thoughts.

So, AC recommends accepting this paper.

**Additional Comments On Reviewer Discussion:**

EGZg raised concerns including insufficient implementation details and more experimental results including meta-token, cache optimization, and context length. (6)
nHjx pointed out the limited novelty and unclear efficacy of the proposed head hybrid approach. (5)
wExb raised more discussions with linear Transformer models and InfiniteTransformer. (6)
The authors conducted all the experiments for the concerns raised by three reviewers, all reviewres raised their scores: 6 -> 8, 5 -> 6, and 6 -> 8.

---

### Decision · Program_Chairs · 2025-01-22

Accept (Spotlight)